# Characterizing the Impacts of Semi-supervised Learning for Programmatic Weak Supervision

**Jeffrey Li**[1], **Jieyu Zhang**[1], **Ludwig Schmidt**[1], **Alexander Ratner**[1,2]
[1]University of Washington, [2]Snorkel AI
{jwl2162, jieyuz2, schmidt, ajratner}@cs.washington.edu

## Abstract

Labeling training data is a critical and expensive step in producing high accuracy ML models, whether training from scratch or fine-tuning. To make labeling more efficient, two major approaches are programmatic weak supervision (WS) and semi-supervised learning (SSL). More recent works have either explicitly or implicitly used techniques at their intersection, but in various complex and ad hoc ways. In this work, we define a simple, modular design space to study the use of SSL techniques for WS more systematically. Surprisingly, we find that fairly simple methods from our design space match the performance of more complex state-of-the-art methods, averaging a 3 p.p. increase in accuracy/F1-score across 8 standard WS benchmarks. Further, we provide practical guidance on when different components are worth their added complexity and training costs. Contrary to current understanding, we find SSL is *not* necessary to obtain the best performance on most *existing* WS benchmarks but is more effective when: (1) end models are smaller, and (2) WS provides labels for only a small portion of training examples.

## 1 Introduction

Learning with limited labels is a fundamental challenge in machine learning (ML) applications [12, 35]. To address the significant costs of hand-labeling training sets, *programmatic weak supervision (WS)* has emerged as a high impact research area [25, 31], where the aim is to learn from multiple cheaper sources of *noisy labels*. Meanwhile, *semi-supervised learning (SSL)* is a more classical direction with similar high-level motivations. Instead of generating larger quantities of noisy labels, SSL aims to directly leverage additional *unlabeled data*. It seems natural that these two fields could be applied productively with one another, yet their intersection has not been systematically studied.

In this work, we anchor on WS approaches and study whether they can be enhanced using techniques from SSL. At a high-level, most WS methods consist of two steps. First, a *label model* aggregates a set of weak label sources to noisily label a training set. Commonly, these sources take the form of user-written heuristics (e.g., for sentiment analysis, a user may check for the keyword "great" to provide the label "positive"). Second, an *end model* is learned on this training set. Crucially, weak sources can often abstain (e.g., the absence of "great" may not imply "negative"), leaving certain examples to remain unlabeled and thereby unused. This presents a natural opportunity to use SSL.

Indeed, this approach of using SSL in WS settings has motivated several recent methods [30, 26, 11, 9]. Though these works often attribute their observed improvements to their usage of unlabeled data [33, 30, 26], they also incorporate various algorithmic components in addition to SSL. Thus, we lack clarity about the precise contributions of SSL and whether simpler methods might also suffice. Also, while varying the amount of unlabeled data is crucial when evaluating SSL techniques [22], previous WS benchmarks contain little diversity along this key dimension. Here, we conduct a more systematic study of how useful SSL is in various WS settings, as well as how and when to best employ it.

37th Conference on Neural Information Processing Systems (NeurIPS 2023).

Specifically, we first organize the intersection between SSL and WS by proposing an explicit design space, centered around disentangling the following key methodological considerations:

(1) *Thresholding: What to treat as (un)labeled?* Because WS uses heuristics for labeling, it often can only provide labels for a subset of examples, leaving the rest as unlabeled. Further, it can also be beneficial to *additionally* remove some (likely) incorrect labels provided by WS, as shown by [15]. Since unlabeled data can result in multiple ways when using WS, we view "what to treat as unlabeled" as a non-trivial and first-class axis in our design space.

(2) *SSL Technique: How to use unlabeled examples*? After deciding what data should be treated as unlabeled, one can leverage these examples by simply using any existing SSL technique. Most recent proposals do so via self-training, which uses the end model to periodically provide labels for unlabeled examples; we also try other out-of-the-box SSL methods.

(3) *Re-labeling: Whether to update weak labels during end model training?* Because some labels from WS are incorrect, it can be helpful to *re-label* a training set with the end model during training. This approach, increasingly used by WS methods [30, 11, 4], is often packaged as part of self-training-based SSL. Here, we identify that re-labeling and SSL can be *independently* employed, and we aim to disentangle their impacts.

With our design space, we can organize previous works and modularly generate a variety of methods. We test these methods on several standard WS benchmarks, finding that our design space is sufficient for matching the performance of more complex state-of-the-art methods. We then compare methods within our design space to ablate the importance of each axis and so provide guidance on when each is worth using (and thus more carefully exploring). We summarize our key findings as follows:

- By searching over our design space, we identify two methods that at least match all previous baselines on 6 of 8 WS benchmarks, averaging a 3 p.p. increase in accuracy/F1-score.

- While previous works emphasize utilizing data left unlabeled by WS sources, we find that on 6 of 8 benchmark tasks, SSL is actually *not* necessary for achieving high performance: thresholding and re-labeling can recover 90% of the gains achieved by also using SSL.

- To explain SSL's lack of impact, we find that the small amounts of unlabeled data in these 6 benchmarks (i.e., $< 31\%$) are largely unnecessary for the underlying tasks; when using clean instead of weak labels, ignoring unlabeled examples drops test accuracy by $< 2.5$ p.p.

- In contrast, when WS sources leave more data as unlabeled (i.e., $> 65\%$), SSL is generally worth prioritizing; using it can improve upon thresholding and re-labeling by up to 16 p.p.

## 2 Related Work

**SSL for WS tasks.** Methods in WS have increasingly turned to SSL to improve end model training. This includes DENOISE [26], which incorporates the temporal ensembling SSL algorithm [14], as well as COSINE [30] and KeyClass [9], which both use self-training [16]. Other works [11, 18, 6, 20, 19, 23, 1] apply SSL when learning from weak labels *plus* a small set of clean labels. However, these methods fundamentally differ in their use of SSL, i.e., they define a labeled-unlabeled split based on cleanly versus weakly labeled examples. Further, integrating clean labels into WS enables a greater variety of specialized strategies, so we do not consider this setting in our study. Likewise, [2] assumes additional supervision via heuristics for which examples share the same labels, similar in spirit to consistency-based SSL methods. Finally, we defer a background on SSL in its own right to Section 3.

**SSL for learning with noisy labels.** SSL techniques have been regularly applied in the literature concerning learning from noisy labels [17, 7, 13]. Though this setting is similar to WS, its main difference is that label noise comes from a single "black-box" noising process instead of from multiple explicit WS sources. Thus, the resulting label noise patterns, often also artificially injected in input-independent ways [28], may differ significantly from those in WS. Furthermore, the WS setting contains some examples with no labels since WS sources may abstain.

**Subset selection in WS.** [15] showcases the broad utility of more carefully selecting subsets of weak labels before end model training. In our work, we consider subset selection in the greater context of two other trends in the WS literature, applying SSL and re-labeling. Compared to the core method of [15], we also try a simpler baseline, similar to [9], based on thresholding the existing confidences produced for weak labels. We find that this method offers a competitive alternative on most datasets.

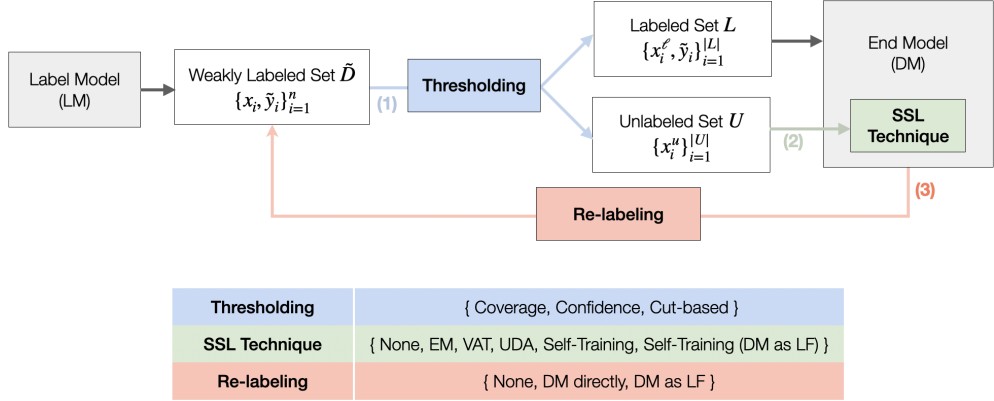

| Thresholding | { Coverage, Confidence, Cut-based } |
|---|---|
| SSL Technique | { None, EM, VAT, UDA, Self-Training, Self-Training (DM as LF) } |
| Re-labeling | { None, DM directly, DM as LF } |

Figure 1: *Overview of our design space.* In WS, a label model produces a weakly labeled training set $\tilde{D}$ which is used to train a discriminative end model. Our design space overlays three decision points on this pipeline: (1) *thresholding*, which filters out some weak labels in $\tilde{D}$ to return labeled and unlabeled sets $L$ and $U$; (2) *SSL technique*, which defines how the end model can still use $U$ during training; (3) *re-labeling*, which uses the end model to update previously used weak labels.

## 3 Design Space

In this section, we first formalize WS and SSL. Then, we describe how our design space overlays on WS, detailing its three key axes along with the specific instantiations of each that we explore in our experiments. Finally, we contextualize which parts of existing work fit within our framework.

### 3.1 Problem Formalization

**Weak supervision (WS)** starts with an unlabeled training set $D = \{x_i\}_{i=1}^n \in \mathcal{X}^n$ drawn from an underlying distribution $(x, y) \sim P$. Labels are provided by a set of labeling functions (LFs) $\{\lambda_j\}_{j=1}^m$, where each $\lambda_j : \mathcal{X} \to \mathcal{Y} \cup \{\emptyset\}$ either labels or abstains (denoted as $\lambda_j(x_i) = \emptyset$) on each $x_i$. The goal is to train a *discriminative end model* (DM) $f : \mathcal{X} \to \mathcal{Y}$ that performs well on $P$. Canonically, WS methods contain two components. First, a *label model* (LM) observes $\left(D, \{\lambda_j\}_{j=1}^m\right)$ and outputs a *weakly labeled* training set $\tilde{D} = \{(x_i, \tilde{y}_i)\}_{i=1}^n$. Second, $f$ is trained using $\tilde{D}$. Often, the LM models $\tilde{y}_i$ probabilistically, allowing $\tilde{y}_i$ to be a soft-label, a vector of probabilities over $\mathcal{Y}$.

Problematically, $\tilde{D}$ may contain several *uncovered* examples, where $\lambda_j(x_i) = \emptyset$ for all $\lambda_j$. Thus, the default practice is then to train $f$ only on *covered* examples that received at least one non-abstaining LF vote. Concurrently, $\tilde{D}$ may also contain several *incorrect* labels, where $\tilde{y}_i \neq y_i$ (or in the soft-label case, $\arg\max_{c \in \mathcal{Y}} \tilde{y}_i[c] \neq y_i$ where $\tilde{y}_i[c]$ is the probability assigned to class $c$). Both issues may lead to sub-optimal performance compared to standard supervised learning. Here, we focus on whether SSL lets us more effectively learn from $\tilde{D}$ in light of these challenges.

**Semi-supervised learning (SSL)** assumes access to a labeled dataset $L = \{(x_i^\ell, y_i)\}_{i=1}^{|L|}$ as well as an unlabeled dataset $U = \{x_i^u\}_{i=1}^{|U|}$. The goal is to obtain a model that performs well on $L$'s underlying distribution despite the limited size of $L$. Though not strictly required, it is often assumed that $|L| << |U|$ and both are drawn from the same test distribution. Generally, each SSL method makes a core assumption about how $P(x)$ relates to $P(y|x)$. Popular categories of methods include *entropy-minimization* [10, 16], which assumes $P(y|x)$ is uncertain only when $P(x)$ is small, and *consistency regularization* [21, 27, 14], which assumes local smoothness of $P(y|x)$ when $x \sim P(x)$.

### 3.2 A Simple Design Space

By considering the standard WS pipeline, we anchor our design space on three natural decision points as shown in Figure 1: (1) thresholding strategy, (2) SSL technique, and (3) whether to re-label previously used weak labels. Importantly, our design space is agnostic to the specifications of label model and end model, though particular choices for each could affect which methods work best.

| Method | Thresholding | SSL Technique | Re-labeling $L$ | Novel LM | Add. Reg. | Add. Labels |
|---|---|---|---|---|---|---|
| Vanilla WS | Coverage | – | – | – | – | – |
| Cutstat [1] | Cut-based (one-time) | – | – | – | – | – |
| Denoise [2] | Coverage | Temp. Ensembling | Self-Train LM | ✓ | – | – |
| Weasel [3] | – | – | Agreement-based | ✓ | – | – |
| Cosine [4] | Confidence (dyn.) | Self-Train | DM directly | – | ✓ | – |
| KeyClass [5] | Confidence (dyn.) | Self-Train | DM directly | – | ✓ | – |
| ASTRA [11] | Coverage | Self-Train (DM as LF) | DM as LF | ✓ | – | ✓ |
| LPA+WL [23] | – | Label Propagation | – | – | – | ✓ |
| SPEAR [18] | – | Entropy Min | Agreement-based | ✓ | ✓ | ✓ |

Table 1: *Contextualizing methods within (middle columns) and outside (rightmost) our design space.* "Novel LM" refers to a method introducing its own label model. "Add. Reg." refers to using additional regularizations, such as contrastive losses and soft-label re-normalization. "Add. Labels" refers to assuming an additional set of clean labels. For methods using clean labels, we do not compare to their results directly, but we can still place elements of their approaches in our design space.

**Thresholding.** To apply SSL, we must first define which examples in $\tilde{D}$ should be considered as part of $L$ and $U$, respectively. Though this is baked into the problem definition in traditional SSL settings, it is a non-trivial choice in WS settings. As a default, standard WS pipelines select $L$ based on labeling function *coverage*, ignoring uncovered examples on which all $\lambda_j$ abstain. However, as shown by [15], removing additional examples from $\tilde{D}$ can help if they are more likely to be incorrectly labeled. In our work, we consider the following strategies for partitioning $\tilde{D}$ into $L$ and $U$:

- *Coverage-based (default):* This removes uncovered points $\{(x_i, \tilde{y}_i) : \lambda_j(x_i) = \emptyset, \forall j\}$.
- *Confidence-based:* Since most label models can output probabilistic labels, a basic yet under-explored approach tried by [9] is to remove examples that have low estimated confidence, here defined as the highest probability assigned to a class. Formally, this removes the examples $\{(x_i, \tilde{y}_i) : \max_{c \in \mathcal{Y}} \tilde{y}_i[c] < \xi\}$ for some threshold $|\mathcal{Y}|^{-1} < \xi < 1$.
- *Cut-based:* This is the method of [15], which at a high-level removes examples whose labels differ most from those of their nearest neighbors in some pre-trained embedding space.

**SSL Techniques.** After partitioning $\tilde{D}$ into $L$ and $U$, we next consider how to still make use of the examples in $U$ to train better end models. By default, WS methods ignore $U$ altogether, but we may instead apply a variety of strategies, such as plugging in any existing SSL technique. Like [22], we limit ourselves to SSL techniques that add unsupervised loss terms during training since these methods tend to achieve the best performance on traditional SSL benchmarks. We also pick representative methods from their taxonomy of approaches:

- *Entropy minimization (EM):* A classical approach, EM [10] penalizes uncertain predictions on unlabeled examples, aiming to place a decision boundary in low-density regions of $P(x)$.
- *Self-training (ST):* Self-training, along with the closely related pseudo-labeling [16, 3], are traditional SSL methods that iteratively use current predictions on unlabeled examples as true labels. Uniquely in the WS setting, we also consider self-training the label model and end model *together*. Inspired by [11], we try one way to do so by feeding the end model as an additional labeling function $\lambda_{m+1}$ used to re-fit the LM. We call this ST (DM as LF).
- *Consistency regularization:* These methods encourage similar predictions on (realistic) per-turbations of unlabeled examples. We use VAT [21] as it does not rely on data augmentations, which are less straightforward for text datasets, the majority of WS tasks. On some tasks, we are also able to use UDA [29], generating perturbations via En-De backtranslations.

**Re-labeling $L$.** Traditional SSL assumes that all labels in $L$ are correct. However, in WS, we may also consider using the end model to correct labels in $L$ as it trains. This increasingly popular technique in WS methods [4, 30, 11, 18] is often packaged with traditional self-training as an overall "SSL method" [30, 11]: instead of using the current model to label just $U$, these methods also do so for $L$. However, this makes it difficult to discern whether these methods improve performance because they leverage $U$ (i.e., apply SSL) or simply because they clean up labels in $L$. Therefore, in our study, we explicitly decouple the use of SSL and re-labeling $L$ as two *independent* decisions; i.e., we can re-label $L$ *regardless* of whether we self-train on $U$, or indeed use any specific SSL technique.

Specifically, we consider two forms of re-labeling analogous to the two types self-training for SSL: (i) using the end model's predictions directly; (ii) using an LM re-fit with the end model as an LF. For

| Datasets | $|\mathcal{Y}|$ | \|Train\| | \|Test\| | # LFs | Coverage | Snork. Precision | Metric |
|---|---|---|---|---|---|---|---|
| IMDb | 2 | 20000 | 2500 | 5 | 87.6% | 74.4% | Acc. |
| Yelp | 2 | 30400 | 3800 | 8 | 82.8% | 75.4% | Acc. |
| Youtube | 2 | 1586 | 250 | 10 | 87.7% | 87.0% | Acc. |
| AgNews | 4 | 96000 | 12000 | 9 | 69.1% | 82.5% | Acc. |
| Trec | 6 | 4965 | 500 | 68 | 95.1% | 60.0% | Acc. |
| Spouse | 2 | 22254 | 2701 | 9 | 25.8% | 65.6% (on Val) | F1 (binary) |
| Chemprot | 10 | 12861 | 1607 | 26 | 85.6% | 58.0% | Acc. |
| Census | 2 | 10083 | 16281 | 83 | 99.1% | 58.0% | F1 (binary) |

Table 2: *Statistics for the WS benchmarks that we use.* Groupings are by task type: text, relation, and tabular classification. Coverage is the percentage of training examples on which at least one LF does not abstain. "Snork. precision" refers to the accuracy of the Snorkel LM on the covered set of inputs.

tractability, however, we only consider re-labeling $L$ in manner (i) when using SSL, except when the SSL technique is already ST (DM as LF). Also, since re-labeling effectively loops the two-stage WS pipeline back onto itself, $N$ rounds of re-labeling could in principle be paired with $N + 1$ separate choices for thresholding and SSL. However, to reduce this search space, we only ever apply the same SSL method across all rounds. For thresholding, we try two possible schedules: *dynamic*, which applies the same thresholding after each re-labeling round and *one-time*, which fixes the $L/U$ split for all rounds of end model training. Finally, some recent methods re-label by using a specialized LM that can be jointly learned with the end model [4, 18]. We do not include this type of *agreement-based* re-labeling in our study, instead focusing on methods agnostic to the form of LM.

### 3.3 Contextualizing previous works

With our design space, we can contextualize several recent WS methods, as shown in Table 1. Overall, this table demonstrates the lack of systematic exploration. For instance, few works perform any thresholding beyond coverage-based, and only [15] and [9] threshold initial LM outputs. Furthermore, assessing the impact of SSL in WS settings is muddled because methods often incorporate several different techniques. A salient example is that COSINE [30] was found by WRENCH [33] to obtain state-of-the-art performance, with both works championing the usage of unlabeled data as a key driver of improvements; however, what COSINE refers to as self-training actually involves pseudo-labeling the *whole* dataset, thereby performing *both* SSL and re-labeling. Further, many methods use techniques outside of our design space entirely, such as novel LMs [26, 4, 11, 18], contrastive learning [30], and soft-label re-normalization [30, 9]. As a result, it remains unclear whether our three axes are necessary or even sufficient to achieve optimal performance. Though our design space is by no means exhaustive, we believe it offers a useful starting point to answer such questions.

## 4 Results

We begin by describing our experiment setup and various baselines in Section 4.1. Then, Section 4.2 explores how our design space yields methods that perform at least as well as the aforementioned baselines. In Section 4.3 and Section 4.4, we conduct extensive ablations on our three axes, finding that SSL is surprisingly unnecessary on most existing WS benchmarks. Finally, Section 4.5 explains this phenomenon and explores settings in which SSL is more helpful.

### 4.1 Experiment Setup

**Datasets and models.** We use 8 classification datasets (see Table 2) and largely follow the end model configurations from WRENCH [33] with a few changes to the hyperparameter grid (Appendix A). For NLP tasks, we both fine-tune RoBERTa pre-trained models and train MLP classification heads on (frozen) RoBERTa embeddings, deferring all results for the latter to the appendix. For tabular tasks, we just train MLPs on raw features. We tune all methods on a shared hyperparameter budget of 50 trials for full RoBERTa fine-tuning and 300 trials for MLPs. All reported test performances are then averages over over three additional test runs, while all error bars are the standard deviations over these runs. Finally, though our design space is compatible with any LM, we use the soft-labels produced by the Snorkel LM from [24]. However, we test robustness to this choice by also trying Majority Voting when comparing with existing methods. From the conclusions of [33], these were most consistently the best LMs across many benchmark tasks.

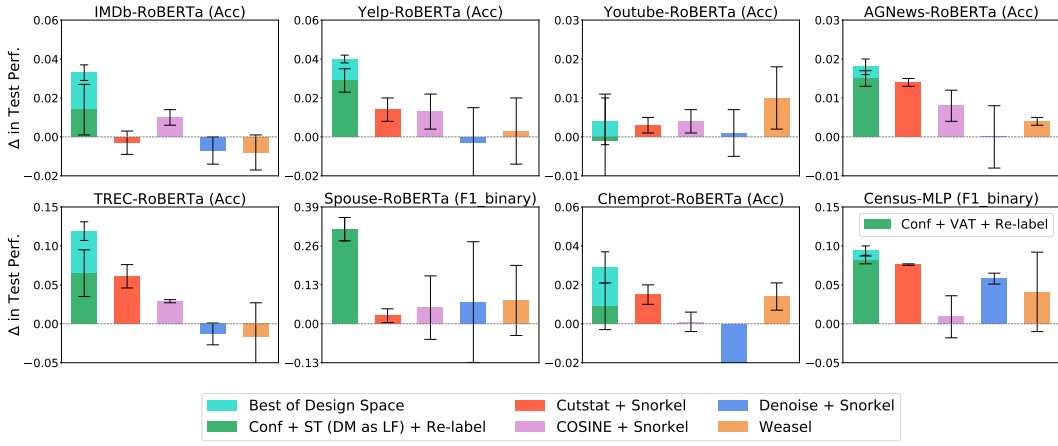

Figure 2: *Test performance for LM = Snorkel.* We compare the best *single* method found (for each end model) from our design space (green) with the best methods found *per-dataset* (turquoise, narrower error caps) as well as four recent methods from the literature. All performances are plotted relative to that of vanilla training (deferring absolute metrics to Table 6 in Appendix B). Note that for Census, a tabular task, the best single method is different since we report the one we found for MLPs.

**Baselines.** We consider five baselines, corresponding to the WRENCH implementations of existing state-of-the-art methods. As Table 1 shows, each incorporates some aspects of our design space, while not necessarily being contained completely within it. We list these methods as follows:

*Vanilla*: applies the default WS pipeline, using coverage thresholding and no SSL or re-labeling.

*Cutstat + {Snorkel, MV}*: uses the thresholding method from [15], with no alterations. Notably, this method *is* contained within our design space as a pure thresholding method.

*COSINE + {Snorkel, MV}*: modifies [30] in two ways when appropriate for fair comparison; the LM used is Snorkel instead of MV except as shown in Table 7, and the DM is an MLP in Table 8.

*Denoise + {Snorkel, MV}*: applies [26] except using Snorkel instead of MV to initialize their label aggregator as shown in Table 7. Also, we use RoBERTa-based end models instead of BERT.

*Weasel*: applies the method from [4] except to train our RoBERTa-based end models for text datasets.

## 4.2 Does our design space yield competitive methods?

We first demonstrate that simple methods from our design space can at least match the performance of all previous methods. When the end model is RoBERTa, our best *single method*, shown in Figure 2, is the combination of (dynamic) confidence thresholding, ST (DM as LF), and re-labeling. This method at least matches the other baselines on 5 of the 7 tasks. On *Chemprot*, only *Cutstat* results in a better point estimate. However, we may also swap in cut-based thresholding into our method, which results a state-of-the-art accuracy of 0.601 (0.008) (as shown in Appendix B, this method is similarly strong when the LM is instead Majority Voting). Finally, Figure 2 illustrates that searching our design space exhaustively *per dataset*–while not necessarily practical–can be even more powerful, improving RoBERTa fine-tuning performance by 1.6 p.p. on average and closing 16% of the remaining gap between our baselines and fully supervised training. Significantly, our methods are strong despite not using additional techniques employed by recent works (as shown in Table 1). This provides empirical justification for focusing only on methods from within our design space in subsequent analyses.

## 4.3 Which axes are most important on current benchmarks?

Though our design space is *sufficient* for strong performance, we now show that SSL is, by and large, not necessary on current benchmarks. Specifically, we conduct an extensive ablation comparing all eight possible subsets of including (or ignoring) each axis, presenting the results in Tables 3 and 9. As an example, for the row "Thresh + SSL," we run every combination within the cross-product of

| Method | IMDb | Yelp | Youtube | AGNews | TREC | Spouse | Chem. | Mean | w/o Spouse |
|---|---|---|---|---|---|---|---|---|---|
| Vanilla | 0.874 (0.002) | 0.925 (0.017) | **0.948 (0.009)** | 0.867 (0.008) | 0.646 (0.011) | 0.215 (0.012) | 0.572 (0.002) | 0.721 (0.004) | 0.805 (0.004) |
| Thresh Alone | 0.885 (0.002) | 0.946 (0.006) | **0.951 (0.002)** | **0.882 (0.001)** | 0.706 (0.015) | 0.263 (0.071) | 0.587 (0.005) | 0.746 (0.010) | 0.826 (0.003) |
| SSL Alone | 0.885 (0.002) | 0.941 (0.008) | **0.955 (0.005)** | **0.885 (0.002)** | 0.660 (0.074) | 0.343 (0.075) | **0.593 (0.010)** | 0.752 (0.015) | 0.820 (0.013) |
| Re-label Alone | **0.901 (0.013)** | 0.944 (0.004) | **0.952 (0.006)** | 0.877 (0.002) | 0.700 (0.014) | 0.332 (0.105) | 0.577 (0.010) | 0.755 (0.015) | 0.825 (0.004) |
| Thresh + SSL | 0.885 (0.018) | 0.959 (0.002) | **0.943 (0.008)** | **0.887 (0.005)** | 0.732 (0.014) | **0.529 (0.057)** | **0.597 (0.009)** | 0.790 (0.009) | 0.834 (0.004) |
| Thresh + Re-label | **0.905 (0.011)** | **0.961 (0.002)** | **0.952 (0.006)** | 0.883 (0.005) | 0.748 (0.023) | 0.236 (0.085) | **0.600 (0.006)** | 0.755 (0.013) | 0.841 (0.005) |
| SSL + Re-label | 0.894 (0.008) | 0.953 (0.008) | **0.951 (0.002)** | 0.882 (0.001) | 0.702 (0.035) | 0.365 (0.073) | 0.583 (0.003) | 0.761 (0.012) | 0.828 (0.006) |
| Thresh + SSL + Re-label | **0.907 (0.004)** | **0.965 (0.002)** | 0.949 (0.010) | 0.885 (0.003) | **0.765 (0.012)** | **0.531 (0.039)** | **0.601 (0.008)** | 0.800 (0.006) | 0.845 (0.003) |
| Entire Design Space | **0.907 (0.004)** | **0.965 (0.002)** | **0.952 (0.006)** | 0.885 (0.002) | 0.765 (0.012) | 0.531 (0.039) | 0.601 (0.008) | 0.801 (0.006) | 0.846 (0.003) |
| Fully supervised | 0.932 (0.005) | 0.976 (0.001) | 0.967 (0.013) | 0.918 (0.006) | 0.966 (0.004) | – – | 0.894 (0.012) | – – | 0.943 (0.003) |

Table 3: *Axis ablation for LM = Snorkel, DM = RoBERTa.* Each entry corresponds to picking a specific method within our design space using validation tuning. Blue numbers are the highest for a given dataset, while bold numbers are those within error bars of the best result.

all *non-default* thresholding and SSL techniques (i.e., {Conf-based, Cut-based} × {EM, VAT, UDA, ST, ST w/ DM}) and then select the best of these combinations using the validation set. For "Entire Design Space," we perform this method selection procedure over all methods from the design space.[1]

From this analysis, we observe that SSL helps only in limited scenarios. Including SSL on *Spouse* yields a 0.2 increase in F1-score compared to not using any form of SSL. However, on all the other six tasks, "Thresh + Re-label" performs at least within a standard deviation of the best method, making up 90% of the gap between "Vanilla" and "Thresh + SSL + Re-label." One explanation for this result is the distinctly lower coverage LF set for *Spouse*, a factor we explore in depth in Sec 4.5. Further, model size may also play a role. In the corresponding Table 9 (Appendix B) for MLPs, "Thresh + Re-label" can make up only 68% of the gap between "Vanilla" and "Thresh + SSL + Re-label."

## 4.4 What are the best instantiations of each axis?

Having compared the three axes at a macro-level, we now zoom in on each. Specifically, we compare all implementations of a given axis when paired with the best possible setting of the other two. For thresholding and SSL, we find that simple or previously underexplored approaches are worth using.

**Thresholding.** Examining Table 4, we see that thresholding is significantly helpful in most cases. This extends the conclusions of [15], showing that removing labels still helps *even when also using SSL and re-labeling*. Interestingly, the simpler confidence-based threshold matches the cut-based method (within error bars) on all datasets except *Chemprot*.

| | IMDb | Yelp | AGNews | TREC | Spouse | Chem. |
|---|---|---|---|---|---|---|
| Cov | **0.901 (0.013)** | 0.953 (0.008) | **0.885 (0.002)** | 0.702 (0.035) | 0.365 (0.073) | 0.593 (0.010) |
| Cut | **0.907 (0.004)** | **0.965 (0.000)** | **0.887 (0.005)** | **0.765 (0.012)** | 0.531 (0.007) | **0.605 (0.002)** |
| Conf | 0.906 (0.005) | **0.965 (0.002)** | 0.884 (0.002) | 0.741 (0.029) | **0.531 (0.039)** | 0.593 (0.010) |

Table 4: *Closer look at thresholding for DM = RoBERTa.* We report the best performance of any method that uses each type of thresholding. Blue indicates the best result while bold indicates being within error bars of the best.

**SSL and Re-labeling.** For these two axes, we observe largely similar trends in Tables 10 and 11 in Appendix C.2. SSL and re-labeling are each only strictly necessary for a minority of datasets, outperforming "no SSL" and "no re-labeling" (beyond error bars) on just one and two datasets, respectively. As such, all SSL methods also tend to perform similarly. For MLP models, though, where SSL is more useful, we see from Table 14 that VAT and ST (DM as LF) are the most consistent. No other method comes within error bars on four datasets.

---

[1]This differs from "Thresh + SSL + Re-label" because methods for that row must employ *non-default* choices.

## 4.5 When is SSL more useful?

To explain why SSL is largely redundant on most existing WS benchmarks, we show that the unlabeled data in each task is generally unnecessary for learning strong models. However, when using lower coverage LF sets, a setting not captured by these benchmarks, ignoring unlabeled data can significantly compromise performance, allowing SSL more room to be impactful.

**Data gaps in WS.** We can think of any WS training set $L = \{(x_i^\ell, \tilde{y}_i)\}_{i=1}^{|L|} \subseteq \tilde{D}$ as suffering from:

(1) *Limited size:* Since not all examples are labeled, the quantity of labels $|L|$ may be insufficient.
(2) *Coverage bias:* Since LFs abstain based on feature-dependent rules, the inputs in $L$ represent a biased subpopulation of the true distribution that the test set is drawn from.
(3) *Label noise:* Since LFs are just heuristics, labels $\tilde{y}_i$ can be biased towards incorrect classes.

*We hypothesize that SSL adds more unique value within our design space when gaps (1) and (2) are more significant.* When gap (3) is the only dominating factor, there is less reason to expect SSL to outperform thresholding and re-labeling since the latter two more directly attempt to address label noise. However, these two axes still do not use the unlabeled examples, which cause gaps (1) and (2).

**Label noise (not unlabeled data) is the main gap on existing benchmarks.**

To measure the relative importance of the three data gaps, we compare the following models:

*GT (Cov)*: the model trained on the ground-truth labels for only LF-covered inputs, removing gap (3) but retaining gaps (1) and (2).

*GT (Full)*: the model trained with a clean and fully labeled version of $D$, removing all gaps.

Given our hypothesis, we would expect these models to perform *similarly* on WRENCH benchmarks. Indeed, as shown in Figure 3, the largest gap between them is < 2.5 p.p.[2]

**SSL helps more when coverage is lower.** While SSL's ineffectiveness on existing benchmarks corresponds to label noise being the predominant data gap, we would ideally also show that SSL is *more* useful when the other two gaps are significant, i.e., when *GT (Cov)* performs markedly worse than *GT (Full)*. One natural way to explore this would be to test on *lower coverage* LF sets, which result in more unla-

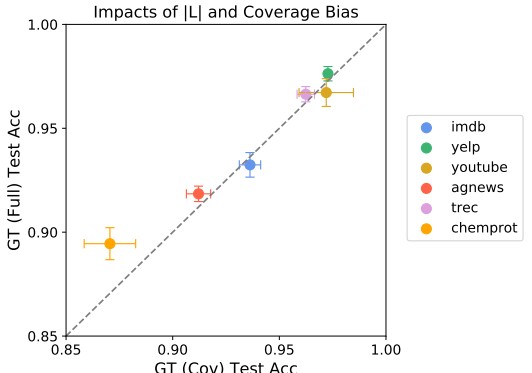

Figure 3: *Measuring the importance of unlabeled data on WS benchmarks.* Assuming no label noise, removing unlabeled data for *GT (Cov)* decreases performance minimally compared to *GT (Full)*.

beled data. However, most existing benchmarks have high coverage, often above 80% as shown in Table 2. To overcome this limitation, we first explore a wider range of coverage levels by subsampling or generating LFs on existing datasets. We also create two new WS text classification tasks based on publicly avaialble datasets, *Massive18* [8] and *Banking77* [5]; these tasks have larger label spaces than all WRENCH tasks and therefore require more effort (i.e., LFs) to obtain high coverage. Finally, we explore the less frequently studied tabular setting, using *Mushroom*, *Spambase*, and *PhishingWebsites* in a setup similar to that of [32]. We defer details on all these data/LF sets to Appendix D.

From this analysis, we first see that smaller coverage levels (within the ranges we test) do not always cause *GT (Cov)* to perform poorly. On some tasks (top two rows of Figure 4), including the three tabular and our two new ones, *GT (Cov)* performs significantly worse as coverage decreases.[3] In contrast, on *IMDb*, *Yelp*, and *AGNews* (bottom row of Figure 4), *GT (Cov)* surprisingly comes within 2% of *GT (Full)* even when coverage drops to 10-20%.

Importantly, the partitioning of datasets based on the performance of *GT (Cov)* also corresponds to the effectiveness of using SSL over not using it. For each LF set used, we run the methods within

---

[2]Unfortunately, *Spouse* does not have ground-truth training labels, preventing us from adding it to Figure 3.
[3]In Appendix E, we show that coverage bias (not limited size) is the main cause of these performance drops.

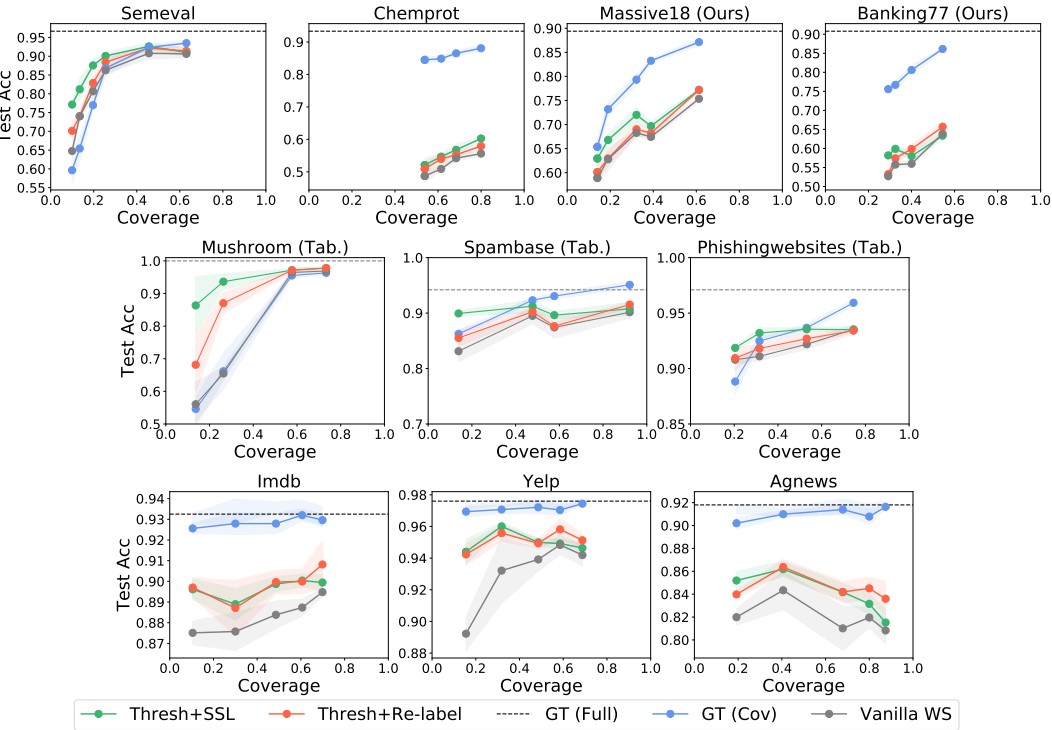

Figure 4: *Impacts of coverage level on data gaps and the effectiveness of SSL.* Plotting *GT (Full)* (gray-dashed) and *GT (Cov)* (blue), we measure the impacts of removing label noise across nested LF subsets. We also compare the effectiveness of Thresh + SSL (green) to Thresh + Re-label (red). For datasets more impacted by coverage bias and limited size (larger gaps between dashed and blue lines), SSL adds more unique value. "(Ours)" refers to datsets we introduce. "(Tab.)" refers to tabular tasks.

a reduced version of our design space: we allow for confidence thresholding and try both versions of self-training to perform SSL (i.e., by labeling points in $U$) or to re-label (i.e., by labeling points in $L$). For the tabular tasks, we also try VAT for the SSL technique because of its effectiveness for MLPs. We plot the best performing methods for both "Thresh + SSL" and "Thresh + Re-label" in Figure 4.[4] As shown in the top two rows, "Thresh + SSL" is enough to consistently outperform "Thresh + Re-label" at lower coverage levels. In contrast, in the bottom row, where *GT (Cov)* drops in performance by < 2 p.p., "Thresh + SSL" continues to perform within errors of "Thresh + Re-label."

Overall, these results suggest that SSL can indeed be useful in WS settings that differ from those captured by standard benchmarks. At lower coverage levels (i.e., <35%), SSL is consistently worth trying; in practice, SSL may allow users to reduce the number of LFs they need to write in order to achieve a particular target performance. On our tabular tasks, these savings seem especially impactful; SSL allows one to achieve performance equivalent to having all 40 LFs (highest coverage plotted) with at most 20 LFs (second lowest coverage plotted).

## 5   Conclusion

We constructed a design space for combining SSL and WS, using it to both contextualize and match the performance of state-of-the-art WS methods. We show that on existing WS benchmarks, using the unlabeled data is surprisingly not essential. However, SSL can be more useful when training MLPs and when the LFs cover fewer examples. Some limitations of our work include: (1) a lack of finer-grained heuristics for more efficiently navigating our design space when given a new task; (2) relying on ground-truth labels to measure the relative importance of different data gaps; (3) using automated ways to generate low coverage LF sets which may not fully represent those written by real users. We see all of these limitations as seeding important directions for future work.

---

[4]Unlike in Section 4.3, here we also allow "Thresh" in "Thresh + {SSL, Re-label}" to be coverage-based.

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

# A Experiment Details

## A.1 Search Spaces

| Method | Hyperparameters | Description | Range |
|---|---|---|---|
| Snorkel LM | lr | learning rate | 1e-5,1e-4,1e-3,1e-2,1e-1 |
| | weight_decay | weight decay | 1e-5,1e-4,1e-3,1e-2,1e-1 |
| | num_epoch | the number of training epochs | 5,10,50,100,200 |
| MLP | batch_size | batch size | 32,128,512 |
| | lr | learning rate | 1e-4,1e-3,1e-2 |
| | dropout | dropout probability | 0.2, 0.0 |
| | weight_decay | weight decay | 0.0 |
| | num_layer | the number of hidden layers | 2 |
| | hidden_size | the hidden size of MLP layers | 128, 256, 512 |
| BERT | batch_size | batch size | 32 |
| | lr | learning rate | 1e-5,3e-5,5e-5 |
| | weight_decay | weight decay | 1e-4 |
| COSINE | $T$ | period for updating pseudo-labels | 50,100,200 |
| | $\xi$ | confidence threshold | 0.1, 0.3, 0.5, 0.7, 0.9 |
| | $\lambda$ | weight for confidence regularization | 0.01,0.05,0.1 |
| | $\mu$ | weight for contrastive regularization | 1 |
| | $\gamma$ | margin for contrastive regularization | 1 |
| Denoise | $\alpha$ | momentum term for temporal ensembling | 0.6 |
| | d | size of hidden layer | $2^6 - 2^9$ |
| | c1 | coefficient of denoiser loss | 0.1,0.3,0.5,0.7,0.9 |
| | c2 | coefficient of classifier loss | 0.1,0.3,0.5,0.7,0.9 |
| | c3 | coefficient of unsupervised self-training loss | 1-c2-c1 |
| WeaSEL | $\gamma$ | temperature | 1.0, 0.33 |
| | d | size of hidden layer | $2^6 - 2^9$ |
| | dropout | dropout prob | 0.3 |
| Confidence Thresh | threshold | minimum confidence to keep | np.linspace($\frac{1}{\vert\mathcal{Y}\vert}$ + 0.1, 0.9, 9) |
| Cutstat Thresh | percentile | what percentile as cut-off | np.linspace(0.1, 0.9, 9) |
| Self-Training | $T$ | period for updating pseudo-labels | 50, 100 |
| Self-Train (DM as LF) | $T$ | period for updating pseudo-labels | 50, 100 |
| EM | $\lambda$ | the coefficient for EM loss | 1, 0.5, 1e-1, 1e-2, 1e-3 |
| VAT | $\lambda$ | the coefficient for VAT unsupervised loss | 1, 0.5, 1e-1, 1e-2, 1e-3 |
| | VAT ip | Iterations of the power method for VAT | 1 |
| | $\xi$ | Finite difference for approximation in VAT | 0.05, 0.1, 0.5, 1, 5 |
| | $\epsilon$ | VAT Perturbation distance | 1e-6, 1e-3, 1, 2.5, 5 |
| UDA | $\lambda$ | the coefficient for UDA unsupervised loss | 4, 2, 1 0.5, 1e-1 |
| | tsa_sched | schedule for training signal annealing | none, log |
| | $\tau$ | temperature for prediction sharpening | 1, 0.85, 0.4 |

Table 5: Search spaces organized by type of method: label models, end models, baselines, thresholding, and SSL techniques.

## A.2 Implementation Details

Here we provide high-level details on the various components of our design space. For further details, please see our codebase at `https://github.com/jeffreywpli/SSL4WS`.

**Label Models.** Of the two label models that we use, we must tune parameters only for Snorkel. We tune over the search space in Table 5 once per dataset; we then fix the hyperparameters and seed for fitting Snorkel to ensure the same exact weak labels are given to all methods. This seed is chosen from a pool of three, also based on the validation set.

**End Models.** For MLPs, we increase the range of hidden sizes compared to WRENCH and tune for dropout instead of weight decay. For RoBERTa, we do not tune the batch size in order to reduce the search space, observing minimal differences compared to using an alternative batch size of 16. Following WRENCH, we perform early stopping based on validation performance for all methods. Specifically, we use a patience of 1000 steps for MLPs and 100 steps for RoBERTa.

**Thresholding.** Whenever applying *dynamic* versions of thresholding, we use the same threshold value across different rounds of end model training for simplicity. Setting separate thresholds for different stages of learning (or adaptively) could be an interesting direction for future work.

**SSL Techniques.** We make note of some important details relevant to specific SSL techniques:

- For the two self-training methods, we use a training schedule similar to that of COSINE, whereby we divide training into two stages: In the first stage, the end model is trained on weak labels as is done in vanilla WS end model training. In the second stage, new pseudo-labels are generated upon reaching a pre-specified update period of {50,100} steps.

- For self-training, we also allow for applying thresholds to the pseudo-labels given to unlabeled data. When combining this approach with thresholding the initial weak labels, we again use the same threshold value.

- When using ST (DM as LF), we do not re-tune the label model-specific hyperparameters for Snorkel in subsequent self-training rounds. We default to instead using the same hyperparameters found from the initial search.

- For EM, VAT, and UDA, we sample batches with equal numbers of labeled and unlabeled examples when taking each step but tune a weighting parameter to balance their losses.

- When applying VAT to RoBERTa models, we calculate perturbations with respect to the RoBERTa-based features (which are continuous) instead of the raw text inputs.

- We try UDA on the text classification datasets where translations are straightforward to apply (i.e., do not require extra annotation effort). This includes *Youtube*, *Trec*, *AGNews*, and *Yelp*. For *Yelp*, where some examples are too long, we split examples at the sentence-level and backtranslated each sentence before concatenating following [29].[5] For the other text datasets, which are relation classification, we did not use UDA as it requires non-trivially annotating the entity spans in the backtranslations.

**Re-labeling.** For re-labeling, we again use a two-stage training schedule where the second stage contains a label update period (as explained when discussing self-training). When combining re-labeling with thresholding, we consider the set of examples with candidate labels (i.e., which are possibly removed when performing dynamic thresholding) to be fixed after the initial round of thresholding (of LM outputs). Thus, re-labeling provides new pseudo-labels only for examples that survived the initial thresholding. This is so as to cleanly differentiate it from SSL.

**Compute.** We ran all MLP experiments on AWS, using up to four `g4dn.4xlarge` EC2 instances at one time. Each instance allowed for running up to 10 different experiments (i.e., here considered as a hyperparameter sweep for any specific method from our design space) in parallel. For all RoBERTa training runs, we ran our experiments on a fleet of up to 15 NVIDIA-A40 GPUs hosted on the University of Washington Hyak cluster. Each experiment fits on a single GPU, which is large enough to use our batch sizes without needing gradient accumulation.

---

[5]*ImdB* had a similar length issue, but unfortunately the version used in WRENCH had been stripped of punctuation, rendering sentence-splitting non-trivial.

# B  Additional Results for Baseline Comparisons

## B.1  Additional Plots

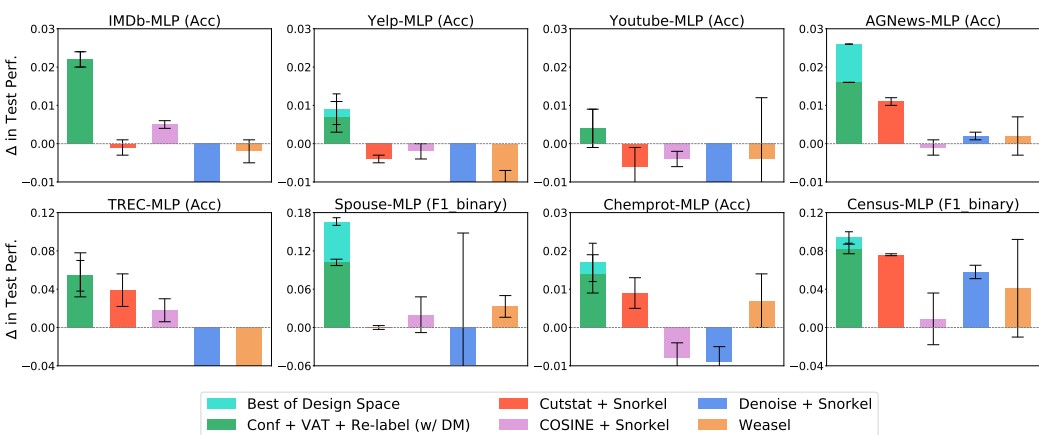

Figure 5: *Test performance for LM = Snorkel, DM = MLP.* We compare a selected method from our design space (green) with four recent proposals from the literature. We plot all performances relative to that of vanilla training (see Table 8 for the absolute metrics).

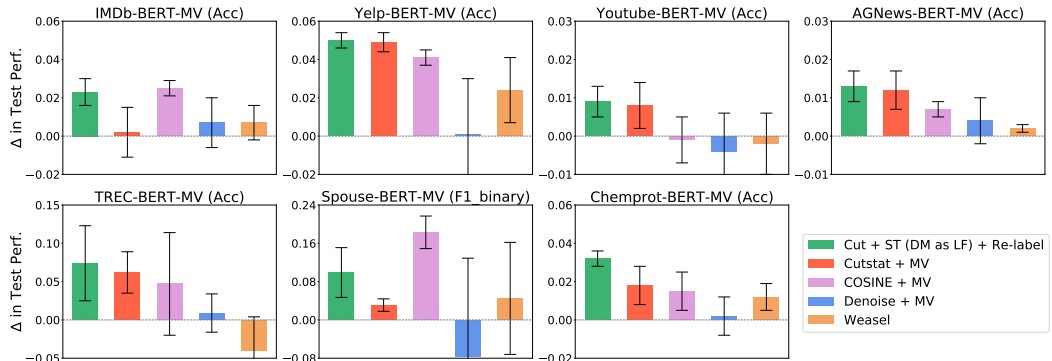

Figure 6: *Test performance for LM = MV, DM = RoBERTa.* We compare a selected method from our design space (green) with four recent proposals from the literature. We plot all performances relative to that of vanilla training (see Table 7 for the absolute metrics). For this setting, we did not run an exhaustive search over the whole design space so we do not report "Best of Design Space."

**MLP Results.** While fully fine-tuning RoBERTa end models provides better overall performance across text-based tasks, we also provide results for MLP end models for completeness. Here, the best single-method across datasets was (one-time) confidence thresholding + VAT + re-labeling. This method achieves the best point estimate on all 8 datasets compared to previous baselines.

**Majority Voting Results.** We also ablate the specific choice of label model to be Majority Voting for the RoBERTa experiments. Here, we mostly use the same method as for when the label model is Snorkel. However, we found that swapping in cut-based thresholding tended to perform better than sticking with confidence-based. This behavior can perhaps be explained by the relatively cruder confidence estimates for Majority Voting soft-labels; instead of learning a probabilistic model, the soft-labels for Majority Voting simply use the ratios of LF votes (e.g. for a binary task, if an example received 2 negative and 3 positive votes, the soft-label would be [0.4, 0.6]). Overall, we find that this method at least comes within error bars of previous baselines on all datasets though the point estimate is 8.4 p.p. lower on *Spouse*. Notably though, the best overall performance on *Spouse* is still obtained by applying our design space on top of the Snorkel label model (see Tables 6 and 7).

## B.2 Tables with Absolute Performance

| Method | IMDb | Yelp | Youtube | AGNews | TREC | Spouse | Chemprot |
|---|---|---|---|---|---|---|---|
| Vanilla | **0.874** (**0.002**) | 0.925 (0.017) | **0.948** (**0.009**) | 0.867 (0.008) | 0.646 (0.011) | 0.215 (0.012) | 0.572 (0.002) |
| Cutstat + Snorkel | **0.871** (**0.006**) | 0.939 (0.006) | **0.951** (**0.002**) | **0.881** (**0.001**) | **0.707** (**0.015**) | 0.242 (0.023) | **0.587** (**0.005**) |
| Conf + ST (DM as LF) + Re-label | **0.888** (**0.013**) | **0.954** (**0.006**) | 0.947 (0.012) | **0.882** (**0.002**) | **0.711** (**0.030**) | **0.531** (**0.039**) | 0.581 (0.012) |
| Best of Design Space | **0.907** (**0.004**) | **0.965** (**0.002**) | 0.952 (0.006) | 0.885 (0.002) | **0.765** (**0.012**) | 0.531 (0.039) | **0.601** (**0.008**) |
| COSINE (RoBERTa) + Snorkel | 0.884 (0.004) | 0.938 (0.009) | 0.952 (**0.003**) | 0.875 (0.004) | 0.675 (0.002) | 0.269 (0.106) | 0.573 (0.005) |
| Denoise (RoBERTa) + Snorkel | 0.867 (0.007) | 0.922 (0.018) | **0.949** (**0.006**) | 0.867 (0.008) | 0.633 (0.014) | 0.287 (0.202) | 0.544 (0.005) |
| Weasel (RoBERTa) | **0.866** (**0.009**) | 0.928 (0.017) | 0.958 (**0.008**) | 0.871 (**0.001**) | 0.629 (0.044) | 0.293 (0.117) | **0.586** (**0.007**) |

Table 6: Test performance for LM = Snorkel, DM = RoBERTa. Blue referes to the best performance of a single method (i.e., excluding "Best of Design Space", which reports the best method *per dataset*) while bold refers to being within standard deviation error bars of the best method. Red indicates where "Best of Design Space" outperforms the best single method outside of error bars.

| Method | IMDb | Yelp | Youtube | AGNews | TREC | Spouse | Chemprot |
|---|---|---|---|---|---|---|---|
| Vanilla | 0.859 (0.004) | 0.904 (0.017) | **0.960** (**0.014**) | 0.869 (0.004) | 0.669 (0.012) | 0.248 (0.030) | 0.574 (0.003) |
| Cutstat + MV | 0.861 (0.013) | **0.953** (**0.005**) | **0.968** (**0.006**) | **0.881** (**0.005**) | **0.731** (**0.027**) | 0.279 (0.013) | 0.592 (0.010) |
| Cutstat + ST (DM as LF) + Re-label | **0.882** (**0.007**) | **0.954** (**0.004**) | **0.969** (**0.004**) | **0.882** (**0.004**) | **0.743** (**0.049**) | **0.347** (**0.052**) | **0.606** (**0.004**) |
| COSINE (RoBERTa) + MV | **0.884** (**0.004**) | 0.945 (0.004) | 0.959 (**0.006**) | 0.876 (**0.002**) | **0.716** (**0.067**) | **0.431** (**0.034**) | 0.589 (0.010) |
| Denoise + MV | 0.866 (0.013) | 0.905 (0.029) | 0.956 (**0.010**) | 0.873 (**0.006**) | **0.678** (**0.025**) | 0.172 (0.205) | 0.576 (0.010) |
| Weasel (RoBERTa) | 0.866 (0.009) | 0.928 (0.017) | 0.958 (**0.008**) | 0.871 (0.001) | 0.629 (0.044) | **0.293** (**0.117**) | 0.586 (0.007) |

Table 7: Test performance for LM = Majority Voting, DM = RoBERTa. Blue referes to the best performance of a single method while bold refers to being within standard deviation error bars of the best method.

| Method | IMDb | Yelp | Youtube | AGNews | TREC | Spouse | Chemprot | Census |
|---|---|---|---|---|---|---|---|---|
| Vanilla | 0.828 (0.001) | 0.917 (0.001) | **0.925** (**0.004**) | 0.854 (0.002) | 0.623 (0.004) | 0.228 (0.013) | 0.544 (0.002) | 0.511 (0.012) |
| Cutstat + Snorkel | 0.827 (0.002) | 0.913 (0.001) | **0.919** (**0.005**) | 0.865 (0.001) | **0.662** (**0.017**) | 0.228 (0.003) | **0.553** (**0.004**) | 0.587 (0.001) |
| Conf + VAT + Re-label | **0.850** (**0.002**) | **0.924** (**0.004**) | **0.929** (**0.005**) | **0.870** (**0.000**) | **0.678** (**0.023**) | **0.330** (**0.005**) | **0.558** (**0.005**) | **0.593** (**0.005**) |
| Best of Design Space | **0.850** (**0.002**) | 0.926 (0.004) | 0.929 (0.005) | **0.880** (**0.000**) | 0.677 (0.016) | **0.394** (**0.006**) | 0.561 (0.005) | **0.605** (**0.006**) |
| COSINE (MLP) + Snorkel | 0.833 (0.001) | 0.915 (0.002) | **0.921** (**0.002**) | 0.853 (0.002) | 0.641 (0.012) | 0.248 (0.028) | 0.536 (0.004) | 0.520 (0.027) |
| Denoise (MLP) + Snorkel | 0.806 (0.011) | 0.895 (0.006) | 0.528 (0.000) | 0.856 (0.001) | 0.568 (0.003) | **0.169** (**0.207**) | 0.535 (0.004) | 0.569 (0.007) |
| Weasel (MLP) | 0.826 (0.003) | 0.905 (0.005) | **0.921** (**0.016**) | 0.856 (0.005) | 0.446 (0.117) | 0.261 (0.017) | **0.551** (**0.007**) | 0.552 (0.051) |

Table 8: Test performance for LM = Snorkel, DM = MLP. Color-coding is done in the same way that is described in Table 6.

# C  Additional Ablation Results

## C.1  Axis Ablation for MLP

| Method | IMDb | Yelp | Youtube | AGNews | TREC | Spouse | Chemprot | Census | Mean |
|---|---|---|---|---|---|---|---|---|---|
| Vanilla | 0.828 (0.001) | 0.917 (0.001) | 0.925 (0.004) | 0.854 (0.002) | 0.623 (0.004) | 0.228 (0.013) | 0.544 (0.002) | 0.511 (0.012) | 0.679 (0.002) |
| Thresh Alone | 0.833 (0.002) | 0.924 (0.002) | 0.915 (0.002) | 0.865 (0.001) | **0.678** (0.027) | 0.228 (0.003) | 0.547 (0.004) | 0.587 (0.001) | 0.697 (0.003) |
| SSL Alone | 0.841 (0.004) | 0.918 (0.002) | **0.932** (0.000) | 0.877 (0.001) | 0.605 (0.008) | **0.375** (0.029) | 0.533 (0.003) | 0.518 (0.015) | 0.700 (0.004) |
| Re-label Alone | 0.833 (0.001) | 0.924 (0.002) | **0.917** (0.014) | 0.862 (0.001) | 0.631 (0.024) | 0.243 (0.003) | 0.549 (0.007) | 0.526 (0.027) | 0.686 (0.005) |
| Thresh + SSL | 0.838 (0.004) | **0.926** (0.004) | 0.929 (0.002) | 0.878 (0.001) | **0.691** (0.012) | **0.394** (0.006) | 0.560 (0.001) | **0.605** (0.006) | 0.728 (0.002) |
| Thresh + Re-label | 0.837 (0.002) | 0.925 (0.003) | 0.917 (0.007) | 0.870 (0.001) | **0.677** (0.016) | 0.270 (0.007) | **0.558** (0.002) | 0.602 (0.002) | 0.707 (0.002) |
| SSL + Re-label | 0.835 (0.001) | **0.925** (0.006) | **0.935** (0.005) | 0.875 (0.002) | 0.662 (0.011) | **0.347** (0.043) | 0.557 (0.004) | 0.542 (0.025) | 0.710 (0.006) |
| Thresh + SSL + Re-label | **0.850** (0.002) | **0.930** (0.001) | 0.929 (0.005) | **0.880** (0.000) | 0.671 (0.034) | 0.347 (0.043) | **0.561** (0.005) | **0.606** (0.001) | 0.722 (0.007) |
| Entire Design Space | **0.850** (0.002) | **0.926** (0.004) | **0.929** (0.005) | **0.880** (0.000) | **0.677** (0.016) | **0.394** (0.006) | **0.561** (0.005) | **0.605** (0.006) | 0.728 (0.002) |

Table 9: *Axis ablation for LM = Snorkel, DM = MLP.* Each entry corresponds to picking a specific method within our design space using validation tuning. Blue numbers are the highest for a given dataset, while bold numbers are those within error bars of the best result.

## C.2  Axis Instantiations

### C.2.1  RoBERTa

Note that we exclude *Youtube* from these tables because the Vanilla baseline is already within error bars of the best performing methods in Table 6. As such, this dataset would not provide much useful signal for assessing which version of a particular axis is best.

| | IMDb | Yelp | AGNews | TREC | Spouse | Chemprot |
|---|---|---|---|---|---|---|
| No SSL | **0.905** (0.011) | **0.961** (0.002) | **0.883** (0.005) | **0.748** (0.023) | 0.332 (0.105) | **0.600** (0.006) |
| EM | 0.895 (0.005) | 0.958 (0.003) | **0.883** (0.002) | 0.700 (0.018) | 0.286 (0.153) | 0.586 (0.007) |
| ST | **0.907** (0.004) | **0.961** (0.003) | **0.887** (0.005) | **0.765** (0.012) | 0.392 (0.098) | **0.601** (0.005) |
| ST (DM as LF) | 0.889 (0.004) | 0.958 (0.003) | **0.885** (0.003) | 0.725 (0.018) | **0.531** (0.039) | **0.601** (0.008) |
| VAT | **0.900** (0.004) | 0.958 (0.001) | **0.883** (0.002) | **0.743** (0.029) | 0.437 (0.004) | **0.605** (0.002) |
| UDA | – | **0.965** (0.002) | **0.885** (0.003) | **0.753** (0.021) | – | – |

Table 10: *Closer look at SSL for DM = RoBERTa.* We report the best performance of any method that incorporates a given SSL technique.

| | IMDb | Yelp | AGNews | TREC | Spouse | Chemprot |
|---|---|---|---|---|---|---|
| No Re-labeling | **0.885** (0.018) | 0.959 (0.002) | **0.887** (0.005) | 0.732 (0.014) | **0.529** (0.057) | **0.597** (0.009) |
| Re-labeling | **0.907** (0.004) | **0.965** (0.002) | 0.885 (0.003) | **0.765** (0.012) | **0.531** (0.039) | **0.605** (0.002) |

Table 11: *Closer look at re-labeling for DM = RoBERTa.* We report the best performance of any method that either uses or does not use re-labeling.

## C.2.2 MLPs

| Method | IMDb | Yelp | Youtube | AGNews | TREC | Spouse | Chemprot | Census |
|---|---|---|---|---|---|---|---|---|
| Coverage-based | 0.841 (0.004) | **0.925** (**0.006**) | **0.935** (**0.005**) | 0.877 (0.001) | 0.662 (0.011) | 0.375 (0.029) | 0.557 (0.004) | 0.542 (0.025) |
| Cut-based | 0.832 (0.002) | **0.930** (**0.001**) | **0.935** (**0.002**) | **0.880** (**0.000**) | **0.684** (**0.016**) | **0.394** (**0.015**) | **0.566** (**0.003**) | **0.606** (**0.001**) |
| Confidence-based | **0.850** (**0.002**) | **0.928** (**0.002**) | **0.929** (**0.005**) | 0.878 (0.002) | **0.691** (**0.012**) | **0.417** (**0.009**) | **0.560** (0.011) | 0.592 (0.008) |

Table 12: *Closer look at thresholding for DM = MLP.* We report the best performance of any method that incorporates a given thresholding method.

| Method | IMDb | Yelp | Youtube | AGNews | TREC | Spouse | Chemprot | Census |
|---|---|---|---|---|---|---|---|---|
| No SSL | 0.837 (0.002) | 0.927 (0.001) | **0.935** (**0.002**) | 0.870 (0.001) | **0.684** (**0.016**) | 0.270 (0.007) | **0.560** (**0.011**) | 0.602 (0.002) |
| VAT | **0.850** (**0.002**) | 0.926 (0.002) | **0.935** (**0.005**) | **0.880** (**0.000**) | **0.691** (**0.012**) | 0.355 (0.015) | **0.563** (**0.007**) | **0.605** (**0.006**) |
| EM | 0.835 (0.001) | 0.922 (0.001) | 0.925 (0.002) | 0.869 (0.002) | 0.664 (0.007) | 0.268 (0.032) | **0.560** (**0.009**) | 0.595 (0.004) |
| ST | 0.838 (0.001) | **0.928** (**0.002**) | **0.929** (**0.005**) | 0.866 (0.001) | **0.674** (**0.018**) | 0.379 (0.008) | **0.566** (**0.003**) | 0.592 (0.002) |
| ST (DM as LF) | 0.834 (0.001) | **0.930** (**0.001**) | **0.935** (**0.002**) | 0.869 (0.000) | **0.668** (**0.038**) | **0.417** (**0.009**) | 0.561 (0.005) | **0.606** (**0.001**) |

Table 13: *Closer look at SSL for DM = MLP.* We report the best performance of any method that incorporates a given SSL technique.

| Method | IMDb | Yelp | Youtube | AGNews | TREC | Spouse | Chemprot | Census |
|---|---|---|---|---|---|---|---|---|
| No Re-labeling | 0.841 (0.004) | **0.926** (**0.004**) | **0.932** (**0.000**) | 0.878 (0.001) | **0.691** (**0.012**) | **0.417** (**0.009**) | 0.560 (0.001) | **0.605** (**0.006**) |
| Re-labeling | **0.850** (**0.002**) | **0.930** (**0.001**) | **0.935** (**0.005**) | **0.880** (**0.000**) | **0.677** (**0.016**) | 0.347 (0.043) | **0.566** (**0.003**) | **0.606** (**0.001**) |

Table 14: *Closer look at SSL axis for DM = MLP.* We report the best performance of any method that either uses or does not use re-labeling.

# D Beyond standard WS benchmarks: datasets for coverage ablations

## D.1 LF subsampling and procedural generation on existing WRENCH datasets

Overall, WS benchmarks lack agreed-upon ways to explore different LF coverage/precision trade-offs. In this work, we choose to do so by sub-sampling LFs from an overall base set. However, the default LF sets associated with WS benchmarks are not equally as suitable to subsample from. In particular, we are wary about subsampling default LF sets when they:

(1) Contain very few LFs, which reduces the possible granularities of subsampling

(2) Contain LFs which are heterogeneous or have untracked lineages (e.g., $\lambda_1$ closely relates to or was only provided in response to another LF $\lambda_2$).

For datsets where we subsample directly, we choose ones with both larger LF counts and where all LFs are of the same form. This includes:

- *Semeval:* 164 string-matching rules selected by humans from an automated candidate generation procedure [34]
- *Chemprot:* 26 individual keyword-based rules [30]

These contrast with LF sets that are far smaller, more heterogeneous, or have unclear dependencies. For these datasets, we instead procedurally generate LFs using the tools from WRENCH [33]

- *IMDb:* 4 aggregate keyword based rules (i.e., each LF checks for the presence of any of multiple keywords), 1 expression-based rule [26]
- *Yelp:* 7 heuristic rules on keywords, 1 third-party model on polarity of sentiment [26]
- *Agnews:* 9 aggregate keyword based rules split amongst four respective classes [26]

**Subsampling.** On some datasets, we directly subsample the LFs coming from WRENCH, selecting the most accurate LFs in a class-stratified fashion in order to roughly preserve similar ratios of LFs between classes. We also take care not to completely remove all the LFs for a given class, setting the minimum count to 1 per class. We explore subsampling per-class ratios within $[0.1, 0.9]$, avoiding LF sets that are near-duplicates of each other (i.e., having coverage levels within 1% of each other). We subsample based on accuracy to (optimistically) simulate an LF writer who is both careful and has considerable domain expertise; we assume that if they were to end up at a lower coverage LF set (either by writing fewer LFs or by pruning LFs written during a "brainstorming" phase), they would prioritize rules that they are most confident about. Assuming the LF writer has sufficient domain knowledge, these LFs are also the ones more likely to be accurate on the examples they fire on.

**Procedural Generation.** For the datasets where subsampling is less appropriate, we use WRENCH's LF generator to construct LF sets with $\{2, 5, 10, 15, 20\}$ n-gram based LFs per class, choosing the most accurate LFs from the candidate pool that have at least 2% coverage over the training set.

## D.2 New WS benchmarks: Massive18 and Banking77

We also create two new WS benchmarks by writing LFs for the publicly available intent classification datasets *MASSIVE* [8] and *Banking77* [5]. These tasks were chosen a high-level to capture some practical challenges that are not as well-represented by current WRENCH tasks; most notably, they contain significantly higher class counts, which makes them more challenging to write LFs for.

Roughly speaking, assuming one writes uni-polar LFs (i.e., each LF either votes for a single class or abstains), the form of the vast majority of LFs in WRENCH, the number of LFs needed to reach a certain coverage level will likely need to scale with the number of classes.[6] Further, because tasks with larger label spaces require writing more LFs, we also view these tasks as being especially relevant for studying *lower coverage* LF sets. We provide more details about the new tasks as follows:

***Massive18*** is derived from the MASSIVE dataset, a collection of single-shot interactions between human users and general intelligent voice assistants across 52 languages. We exclusively use the

---

[6]This, of course, also assumes that the coverage of LFs within each class does not change dramatically

English-US portion of this dataset and treat the broad *scenarios* (i.e., general domains) as labels instead of the finer-grained intents. For instance, "alarm" is a scenario whereas "alarm_set" and "alarm_remove" are two intents within that scenario. We call the resulting task *Massive18* since there are 18 different scenario classes. Notably, even with this simplification, *Massive18* contains at least nearly double the class count of all previous WRENCH classification tasks (see Table 15 below).

***Banking77*** is a collection of online banking queries along with the corresponding user intents. We use this task as defined with the full set of 77 intents, which can be both quite domain specific and especially fine-grained. If we were to write the *minimum* number of LFs for *Banking77*, i.e. one for each class, this would already require 77 LFs, more than the number of LFs for most previous WRENCH datasets.

| Datasets | $|\mathcal{Y}|$ | $|$Train$|$ | $|$Test$|$ | # LFs | Coverage | Snork. Precision | Metric |
|---|---|---|---|---|---|---|---|
| IMDb | 2 | 20000 | 2500 | 5 | 87.6% | 74.4% | Acc. |
| Yelp | 2 | 30400 | 3800 | 8 | 82.8% | 75.4% | Acc. |
| Youtube | 2 | 1586 | 250 | 10 | 87.7% | 87.0% | Acc. |
| AgNews | 4 | 96000 | 12000 | 9 | 69.1% | 82.5% | Acc. |
| Trec | 6 | 4965 | 500 | 68 | 95.1% | 60.0% | Acc. |
| Massive18 (ours) | 18 | 11514 | 2974 | 59 | 61.3% | 83.8% | Acc. |
| Banking77 (ours) | 77 | 9003 | 3080 | 218 | 54.5% | 76.5% | Acc. |
| Spouse | 2 | 22254 | 2701 | 9 | 25.8% | 65.6% (on Val) | F1 (binary) |
| Chemprot | 10 | 12861 | 1607 | 26 | 85.6% | 58.0% | Acc. |
| Census | 2 | 10083 | 16281 | 83 | 99.1% | 58.0% | F1 (binary) |

Table 15: *Comparing our two new datasets (blue) to previous benchmarks.* Coverage is the percentage of training examples on which at least one LF does not abstain. "Snork. precision" refers to the accuracy of the Snorkel LM on the covered set of inputs.

**Labeling Functions.** For both datasets, we manually write a set of LFs ourselves based upon a randomly sampled *development set* of 250 cleanly-labeled examples. Each LF checks whether a specific keyword (or set of multiple keywords) is a substring of the given example and then votes for a specific class if the substring(s) are present (and otherwise abstaining). We provide some examples of the LFs we wrote in Tables 16 and 17, while we share the full LF sets in our supplied codebase. Importantly, while we believe these LF sets are *reasonable*, they should not be treated as the be-all-end-all and encourage others to iterate or develop their own for these tasks in the future.

Finally, in our experiments, we also try the same subsampling procedure from Appendix D.1 to further explore different coverage levels. For *Massive18* we use the ratios {0.1, 0.5, 0.7, 0.8, 1} and for *Banking77* we use the ratios {0.1, 0.5, 0.7, 1.0}.

| Label | Keyword LFs |
|---|---|
| "alarm" | ["alarm", "wake+up"] |
| "takeaway" | ["takeaway", "delivery", "order"] |
| "social" | ["tweet", "twitter", "facebook", "complain"] |
| "music" | ["what+song", "save+song", "shuffle"] |
| "calendar" | ["calendar", "schedule", "remind"] |

Table 16: *Example LFs for Massive18.* Each row shows one of the possible labels and the associated keyword-based LFs. Note that some LFs contain multiple keywords/substrings that are joined by the "+" character. This signifies that the LF checks whether *all* the keywords supplied are in a given example (though they do not necessarily have to appear in the provided order).

| Label | Keyword LFs |
|---|---|
| "age_limit" | ["age limit", "child", "my son", "my daughter"] |
| "pin_blocked" | ["takeaway", "delivery", "order"] |
| "lost_or_stolen_card" | ["tweet", "twitter", "facebook", "complain"] |
| "verify_my_identity" | ["id+check", "what+ id"] |
| "disposable_card_limits" | ["disposable+limit", "disposable+max"] |

Table 17: *Example LFs for Banking77.* Note the usage of "+", as explained in the caption of Table 16.

### D.3  Tabular Datasets

**Labeling Functions.**  We use three tabular tasks and the LF generation procedure from [32]. At a high-level, this procedure uses the `sklearn` implementation of random forests (i.e., `RandomForestClassifier`) to train multiple decision trees on a small cleanly-labeled development set (i.e., formed by uniformly sampling 5% of the training set in our experiments). Uni-polar LFs are then derived from individual trees. In our experiments, we generate a set of 100 *candidate* LFs for each task and then select the top-{5%, 10%, 15%, 20%} most accurate LFs (per-class) based upon the full training set (i.e., similar in spirit to WRENCH's procedural LF generator). Again, while this selection step isn't possible without access to all the labels, we consider it as a loose (optimistic) approximation of integrating a domain expert's knowledge and judgment.

**Robustness to seeding.** One caveat of this approach for genearting LFs is that the behavior may vary depending on the initial random seed given to `RandomForestClassifier`. To try to account for this, we try multiple seeds and find that the overall conclusions do not change across different runs of the LF generator. As seen in Figure 7, while the absolute performance levels may vary across seeds, SSL is consistently more useful at lower coverage levels.

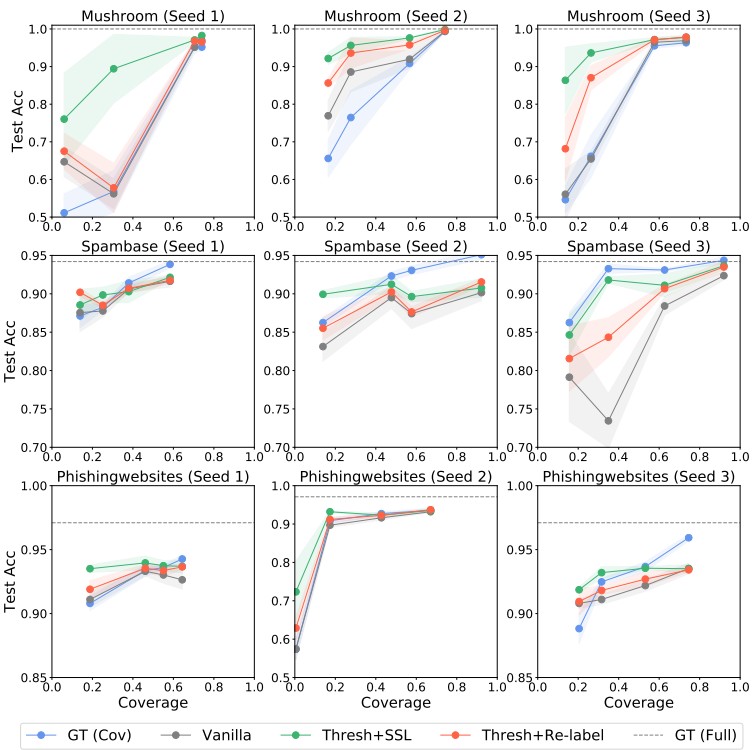

Figure 7: Tabular experiments across different random seeds

# E   Coverage bias explains the value of unlabeled data

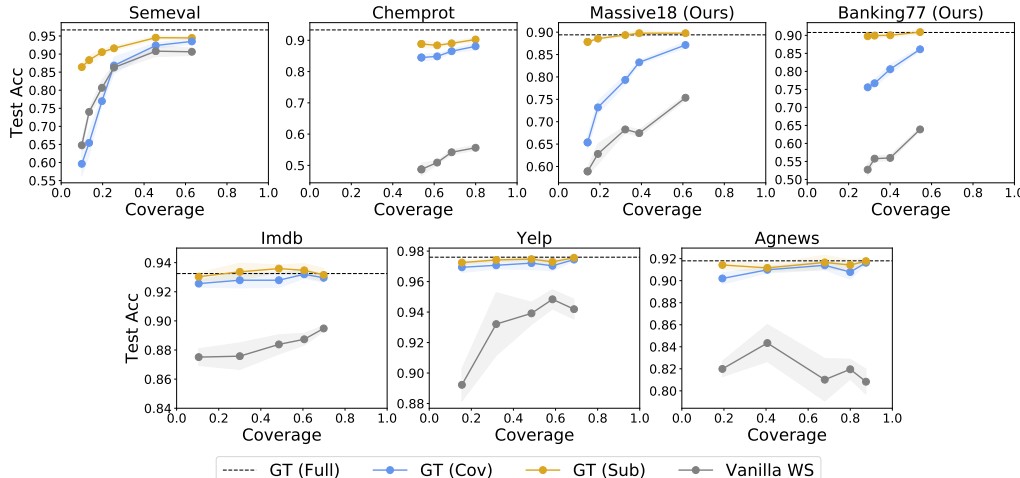

Figure 8: Limited Size versus Coverage Bias. We may assess the relative impacts of the limited size of WS training sets (comparing *GT (Full)* and *GT (Sub)*) versus their coverage bias (comparing *GT (Sub)* and *GT (Cov)*).

In Section 4.5, we observed that on some datasets, as coverage decreases, so does the performance of *GT (Cov)*. Indeed, for any LF set, we can view the gap in performance between *GT (Cov)* and *GT (Full)* as a reflection of the aggregate impact of two *data gaps* that WS datasets suffer from: (1) *limited size* (i.e., not all training examples are labeled) and (2) *coverage bias* (i.e., the examples covered by LFs come from a biased subpopulation of the test distribution). Because *GT (Cov)* simply ignores all unlabeled examples, this gap also then reflects the value of said examples.

A natural follow up question is whether limited size or coverage bias more greatly hinders learning under WS. Here, we further decompose the performance drops we observed between *GT (Cov)* and *GT (Full)* by considering a third model that interpolates between the two. Specifically, we consider *GT (Sub)*, a model that suffers only from limited size but not coverage bias. This model is trained on a training set that shares the same size as the dataset for *GT (Cov)*, but consists of examples *re-sampled uniformly* from the full training set (i.e., eliminating coverage bias).

Once we have trained all three models, we may then simply compare the following performance gaps:

- TestPerf(*GT (Full)*) − TestPerf(*GT (Sub)*): to ablate the impact limited size
- TestPerf(*GT (Sub)*) − TestPerf(*GT (Cov)*): to ablate the impact of coverage bias

As we see in Figure 8, when large gaps exist between *GT (Full)* and *GT (Cov)*, most of this gap is explained by the performance drop between *GT (Sub)* and *GT (Cov)* instead of the drop between *GT (Full)* and *GT (Sub)* (e.g., especially on *Semeval, Massive18, Banking77*). This suggests that coverage bias is far more responsible than limited size, which perhaps makes sense in the context of WS, as LFs can label an arbitrary amount of data but only from the covered set. Also, this means that SSL techniques, initially developed in settings where labeled and unlabeled sets are i.i.d. (i.e., where limited size is the only relevant data gap), actually remain effective in low coverage WS settings despite the significant impacts of coverage bias.

