# OpenReview forum: "Characterizing the Impacts of Semi-supervised Learning for Weak Supervision"
_NeurIPS.cc/2023/Conference — NeurIPS 2023 poster_

### Official Review · Reviewer_xhQz · 2023-06-28

**Soundness:** 4 excellent
**Presentation:** 3 good
**Contribution:** 3 good
**Rating:** 6
**Confidence:** 4

**Summary:**

The authors propose a design space to analyze the effects of WS + SSL, and how to combine them, and on which regimes that SSL help WS. The paper's biggest conclusion is: training using unlabeled datain SSL is mainly unhelpful, except when the main issue is bad labeling function (LF) accuracy. The design space is based on 3 major axis: Thresholding, SSL technique, and re-labelling.

**Strengths:**

- The paper presents a clear framework to analyze the effect of using SSL on WS
- The framework provides a design space that is clear and intuitive, and based on 3 axis that helps t disentangle the effect from WS or SSL
- Important result presented on when SSL is not or is helpful to improve WS

**Weaknesses:**

- Perhaps analysis can also be conducted on snorkel's competitive alternative (e.g., FlyingSquid https://github.com/HazyResearch/flyingsquid)
- While discussing label noise, its important to clarify that WS required label function to have better accuracy than random (ACC > 50%)
- An important ablation and information need also to be included is LF accuracy.
- For the NLP task that uses embedding, its interesting to also analyze how SSL interact with smoothness based methods to improve LF coverage: LIGER https://arxiv.org/pdf/2203.13270.pdf

**Questions:**

See Weakness

**Limitations:**

Yes, limitations is addressed

---

> ### Author Rebuttal · Authors · 2023-08-10
>
> Thanks for your time and helpful feedback! We see your comments as covering two main points, which we will respond to below:
>
> **(W1) Testing on more label models (i.e., FlyingSquid, LIGER)**
>
> **FlyingSquid (FS).** For our experiments, we chose label models (i.e., Snorkel/MajorityVoting) by looking at the results from the WRENCH paper and picking the ones which led to the strongest performance for Vanilla end model training. For FS, we saw its performance was similar in most cases, while considerably worse on several tasks (e.g., Spouse, Chemprot, Trec)– hence we did not prioritize it. Of course, the implicit assumptions here are that: (1) Vanilla performance is also an indicator of performance once our design space is applied on top; (2) our high-level conclusions should translate over across different LMs.
>
> To your point, it could be worth exploring assumptions (1)-(2) further.  As a preliminary experiment, we ran both Vanilla end model training and the method we identified as being the best across datasets (in Sec 4.2) on top of FS labels. From the results in Table 2 in the pdf (attached to global response), the results for FS and Snorkel are extremely similar on Imdb and Yelp. On most other datasets, Snorkel’s better performance for Vanilla translates over to better performance once our method is applied (which still improves upon Vanilla for both Snorkel/FS). The only exception is on Youtube, where FS overtakes Snorkel once our method is applied (though within fairly wide error bars). Of course, more comprehensive verification of (1)-(2) would require running many more methods from our design space on FS. Nonetheless, we believe these results provide some positive initial signal about our assumptions.
>
> **LIGER.** We agree that it would be interesting to apply our analyses on top of this approach given that it directly expands the coverage of LFs. Based on our results, we’d suspect that applying SSL on top of LIGER’s expanded LFs are likely to make less of an impact than on the labels produced by the raw LFs, but it would be nice to verify this!  While we have not yet been able to try such experiments, we would be happy to try including it in future revisions.
>
> **(W2) Discussion/Ablation of Label Noise**
>
> *“The paper's biggest conclusion is: training using unlabeled data in SSL is mainly unhelpful, except when the main issue is bad labeling function (LF) accuracy”*
>
> Just to be on the same page here, we actually see the takeaway as being the other way around: “when the main issue is bad LF accuracy (as opposed to incomplete coverage)”, we find that SSL is less likely to be helpful. Namely, our results in Sec. 4.5 show that on standard WS benchmarks (where SSL is not too helpful),  label noise is the main issue: removing label noise and training only on covered examples allows us to recover the performance of fully supervised learning. However, in settings with lower coverage, label noise is no longer the only significant issue and SSL is indeed more helpful.
>
> *"“It’s important to clarify that WS requires label function to have better accuracy than random”*
>
> Thanks for pointing this out. Especially for theoretical treatments of WS, assumptions about the informativeness of LFs are important and we’d be happy to discuss this more in future revisions. For Snorkel at least, the assumption is that LFs are **on average** better than random. Other approaches may have stronger assumptions though, and we will try to clarify this!
>
> *"“An important ablation and information need also to be included is LF accuracy.”*
>
> We agree that the precisions of LFs are important to consider. In the current manuscript, we do include the precision of the aggregated weak labels (i.e. the label model outputs) for each of the datasets and their full LF sets. In future revisions, we can add this information for each of the reduced coverage setups as well as some summary stats for the individual LFs (similar to Table 5 in the WRENCH paper).
>
> One experiment we’ve started to run since the submission is to try subsampling the least accurate LFs from a candidate pool instead of the most accurate. Unsurprisingly, this results in worse (and arguably less realistic) training sets but does allow us to compare LF sets of similar coverage but different precision levels. From the preliminary results in Figure 1 of the attached pdf, we observe that on Semeval and Massive18, we get qualitatively similar results to the ones we saw before: Thresh+SSL meaningfully outperforms Thresh+Re-label at lower coverage levels but less so at higher ones. In future revisions, we’re happy to extend such an analysis to more datasets!

---

> > ### Comment · Reviewer_xhQz · 2023-08-10
> >
> > Thank you for the clarifications! My initial concerns have been addressed, and I have changed my initial review score

---

### Official Review · Reviewer_1c9G · 2023-06-29

**Soundness:** 3 good
**Presentation:** 3 good
**Contribution:** 3 good
**Rating:** 6
**Confidence:** 4

**Summary:**

This paper studies the impacts of semi-supervised learning (SSL) for programmatic weak supervision (WS) in a systematic way. The authors define a modular design space with three key methodological considerations, thresholding, SSL Technique, re-labeling, to study the use of SSL for WS. Their results show that fairly simple methods from their design space can match the performance of complex state-of-the-art method. Also, SSL is not necessary to obtain the best performance on most WS benchmarks, but is more effective when the end model is small or WS can only label a small portion of training examples.

**Strengths:**

1. The paper provides a systematic design space to study WS. Most of the previous baselines can be placed into the design space which makes a thorough analysis of existing WS methods possible. Also, the paper conducts extensive experiments on multiple design choices and a wide range of datasets - the benchmark results can be beneficial to the community.
2. The best method discovered by the design space can match the performance of more complex methods. This may arouse the interest of practitioners since they can use the proposed design space to find effective WS methods instead of doing a lot of huristic hand-craft.

**Weaknesses:**

1. The findings of the paper are not surprising. Though the authors claim that SSL is not necessary, the experimental results show that SSL can still bring benefits on many datasets - I cannot see why the paper emphasizes SSL is not necessary if it still helps. Another finding of the paper is that SSL is effective when WS can only label a small portion of training examples. This is easy to foresee as SSL can further exploit a large number of unlabeled examples.
2. While this is an analysis paper, all the analyses are empirical and there is no theoretical analysis. Therefore, the conclusions are drawn from the experimental results on the benchmark datasets. It's unsure whether these datasets are representative and extensive enough to cover all cases. One claim made by this work is that "SSL is not necessary to obtain the best performance on most WS benchmarks". A possible reason for this is that the benchmark datasets are rather easy - according to Table 2, most of the datasets are binary classification; for those with more classes, the coverage of WS labeling is very high. I'm skeptical whether this is true for datasets in real application.


**Questions:**

1. This paper involves a lot of acronyms which made it occasionally hard to follow. For "GT" discussed in Section 4.5, does it refer to "Ground Truth"? I didn't find a definition of this acronym in the paper.

**Limitations:**

The authors have already discussed the limitations.

---

> ### Author Rebuttal · Authors · 2023-08-10
>
> Thanks for your time and helpful feedback! To respond to some of your questions and comments:
>
> **(W1a)**  *“The experimental results show that SSL can still bring benefits on many datasets - I cannot see why the paper emphasizes SSL is not necessary if it still helps.”*
>
> We’d like to highlight that on the standard benchmarks, improvements enjoyed by additionally applying SSL mostly do not exceed the stdev error bars when compared to Thresh+Re-label. This is surprising, as we note, because previous works test on these benchmarks and suggest that SSL is a central reason for improved performance. For instance, the WRENCH paper posits (in their Sec. 7) that *“the superiority of COSINE suggests that uncovered data should also be used in training an end model; this inspires [the exploration of] new DM training strategies combined with SSL techniques”.* That being said, in Sec 4.5, our point is exactly that SSL can often be more impactful in scenarios that differ from these standard benchmarks. Though we tried to be explicit about these important nuances in this manuscript, we will work to make them even more upfront in future revisions!
>
> **(W1b)** *“Another finding of the paper is that SSL is effective when WS can only label a small portion of training examples. This is easy to foresee as SSL can further exploit a large number of unlabeled examples.”*
>
> First, we agree that this is not overly surprising, but we still view this as an important practical result given the increasing ubiquity of SSL approaches, e.g. we hope this can provide some guidance for questions such as: “at what coverage levels is SSL useful?” and “is it better to learn from fewer (more-accurate) LFs + SSL or from a larger overall set of LFs?” We also note that there are at least two more non-obvious conclusions when it comes to *high-coverage settings*: (i) even when we allow for thresholding which yields smaller datasets (i.e., more unlabeled data), SSL still helps minimally; and (ii) given we are using programmatic WS, the unlabeled examples, despite their small number, may come from key parts of the input space that are systematically abstained on by LFs; one might expect that SSL could help address this biased coverage of the LFs, but this also does not happen.
>
> **(W3)** *“A possible reason for this is that the benchmark datasets are rather easy - according to Table 2, most of the datasets are binary classification; for those with more classes, the coverage of WS labeling is very high.”*
>
> We agree regarding your assessment of the commonly used datasets in Table 2. Indeed, this skepticism is precisely what motivated us to study WS settings that notably differ from previous benchmarks in Sec 4.5. In particular, we chose to look at:
>
> * LF sets that span a greater range of coverage levels (on the same dataset)
> * Tasks that have considerably larger label spaces (i.e., Massive18, Banking77)
> * Tabular tasks instead of text-based
>
> While we do not claim that these factors now capture all aspects of real applications, our conclusions from these experiments are already that the impactfulness of SSL can indeed change, as you allude to.
>
> **(Q1)** Apologies for any confusions here! GT indeed corresponds to ground-truth in Sec 4.5, which we definitely should take care to explicitly define. We also appreciate the general feedback about acronyms a lot and plan to more carefully reduce our reliance on them in future revisions.

---

> > ### Comment · Reviewer_1c9G · 2023-08-11
> >
> > Thank you for the detailed reply!
> >
> > I agree with the authors' response on the emphasis of Sec 4.5 which provide many interesting insights on when SSL can have benefit. I do think it's beneficial to make this part more upfront in revision. However, I still hold my argument that some findings of the paper are not surprising. I prefer to keep my rating unchanged as it has already shown the inclination of "accept".

---

### Official Review · Reviewer_62vf · 2023-06-30

**Soundness:** 4 excellent
**Presentation:** 4 excellent
**Contribution:** 4 excellent
**Rating:** 8
**Confidence:** 4

**Summary:**

This paper empirically studies the interface between (programmatic) weak supervision (WS) and semi-supervised learning (SSL). Several existing works have tried to leverage SSL techniques and other tricks to improve the performance of weakly supervised learning, since the two settings are fairly similar on the surface.

This paper provides a taxonomy of the key techniques from this line of work, arguing that most methods differ along three main axes: thresholding, SSL technique, and re-labeling. The paper empirically explores combinations across these three axes to study if and when the SSL techniques themselves really improve WS. The best single method of the design space matches or exceeds existing methods from the literature on 5/7 benchmark datasets, indicating that design space is large enough to be representative of the current literature.

Surprisingly, though, the ablation results indicate that the SSL techniques themselves are not really the main cause of good performance in the regime of the most common benchmark datasets (relatively high coverage of weak rules). Instead, thresholding and re-labeling are shown to be responsible for most of the gain of these methods. The SSL techniques are shown to help more when the coverage is (artificially) reduced.

**Strengths:**

- Well-written empirical study that addresses an important question for the WS field, and could influence which directions are explored by future research.

- The key empirical takeaway, that existing SSL methods are mainly helpful in the low-coverage regime, can drive more research in both weak supervision and semi-supervised learning. Many WS papers have been focused on using SSL techniques to improve performance, but the empirical results from this careful study indicate that the main benefits usually come from other tricks like thresholding. This could spur more research into why these other techniques work well and also into new SSL methods that use unlabeled data to improve learning from structured label noise.

- The design space of combinations is empirically shown to be powerful, exceeding or matching the state-of-the-art results on benchmark datasets.

- The WRENCH benchmark datasets tend to have fairly high coverage rates (in my opinion this is one of the few weaknesses of WRENCH). In response, this paper details a carefully thought-out way of artificially inducing lower coverage to study how WS methods perform in this empirically relevant setting, and also provides two new benchmark datasets that have lower coverage rates.



**Weaknesses:**

- Fairly large validation sets are used to select best DM from each training iteration and also to select best combination from the design space. If we have access to (say) 300 labeled examples for validation, can't we use 100 of them for training as well? It's important to note that the authors are following most if not all other works in the WS field with this choice, and the authors appropriately cite works that mix the supervised and weakly-supervised settings. But it would be interesting to see how many of the results remain the same when much smaller (e.g., something like num_classes * 10) validation sets are used. I've found that more complicated methods like DM as LF are less robust in this setting, so that may affect some of the results (but probably not the key takeaway).


**Questions:**



**Limitations:**

The authors appropriately discuss the limitations of their work.

---

> ### Author Rebuttal · Authors · 2023-08-10
>
> Thanks for your time and helpful feedback, we’re glad you appreciated the paper! Regarding your point about validation sets: like you said, we chose to follow the standard setups established by other works, as our main questions were about assessing various trends we saw from the WS literature. However, we agree that these default validation sizes are probably large and it would be valuable to explore the importance of validation data as an additional factor for which methods from our design space succeed/fail. One natural hypothesis would be that as less validation data is used, performance would decrease more sharply for methods that involve more components, i.e., especially the SSL methods since they tend to introduce the most additional hyperparameters. As such, it may be even more difficult for SSL to be usefully applied.

---

### Official Review · Reviewer_vRqg · 2023-07-07

**Soundness:** 4 excellent
**Presentation:** 3 good
**Contribution:** 3 good
**Rating:** 7
**Confidence:** 4

**Summary:**

Getting labeled training data is a bottleneck in the development of machine learning pipelines. Two families of methods that address this are weak supervision (WS), which aggregates multiple sources to produce weak labels, and semi-supervised learning (SSL), which combines a (weakly) labeled dataset and an unlabeled dataset. While there have been works that study how to utilize ideas from WS and SSL to best produce high quality models from unlabeled data, the ways they combine these ideas has been fairly ad-hoc and unstructured. This paper conducts a systematic analysis of how to merge ideas from WS and SSL, breaking it down into three design choices: thresholding (what weakly labeled data to revert to unlabeled), the SSL technique for using unlabeled data, and re-labeling, which creates a self-loop of updating weak labels. The authors find that many existing methods that merge both WS and SSL can be expressed in this design framework, and their design space search finds combinations of strategies (thresholding/ssl/re-labeling) that match existing methods' performance. They study if simpler techniques, such as thresholding based on model confidence, can match more complex techniques. They also uncover that in many cases, the SSL step does not help that much unless there is a significant performance gap due to poor and biased labeling function coverage.

**Strengths:**

Originality:
- This structured survey of ways to use SSL for WS is very novel.

Quality:
- Paper is sound with extensive empirical evaluation. The authors seek to thoroughly understand why they observe that SSL isn't helpful, and design compelling experiments to validate their hypothesis.

Significance:
- I think such a paper is very valuable to those who work on WS/SSL. Personally, I have also thought about how to best use SSL for WS and have not been convinced why one way is better than the other - even trying to prove this theoretically has been technically challenging, as these methods all ultimately use the same amount of information (some labeling functions and an unlabeled dataset; maybe some validation data, etc.) And so perhaps the solution here is not to drill into finding better method but to systematically view these methods as combinations of simpler design choices. This perspective will also be highly useful for practitioners.
- Random thought: These findings on unlabeled data being unnecessary are potentially connected to data pruning literature, where one can discard low confidence points and still have a good model; however, there are cases when these unlabeled points are especially hard and thus worth training on --- maybe this is one perspective to understand why some datasets have a gap between GT-cov and GT-full?

**Weaknesses:**

Clarity:
- Description of methods/baselines were a bit unclear and hard to follow at times. The paper describes these methods only via text and fig 1. It would be helpful to put in the appendix some formalization of these if appropriate; for instance, the two paragraphs on re-labeling L at line 149 took a bit of time to understand (stuff around decoupling SSL and re-labeling, dynamic vs one-time scheduling).

**Questions:**

- Can this framework be applied outside of weak supervision for generic noisy soft labels?
- Seems strange that Massive18, Banking77 and the tabular datasets have this gap from (1) and (2) but Wrench datasets don't. Any idea why?
- In figure 4, why are you comparing thresh+ssl and thresh+re-label? Shouldn't we be studying the marginal benefit of adding SSL?
- See above for suggestions around clarity.

Rebuttal acknowledgement: I have read and acknowledged the author rebuttal. It has addressed my concerns above.

**Limitations:**

Limitations are addressed.

---

> ### Author Rebuttal · Authors · 2023-08-10
>
> Thanks for your time and helpful feedback, we’re glad you appreciated the paper! To respond to your questions and comments:
>
> **Clarity.** We really appreciate your feedback here. In future revisions, we will take care to make these descriptions more explicit in the Appendix.
>
> **Connections to example difficulty.** Interesting thought! Intuitively, using (human-written) labeling functions could risk bias towards labeling “easy points” instead of  “hard points” given that they often take the form of simple rules. Indeed, our results in Appendix E more generally suggest that the gaps between GT (Cov) and GT (Full) are due to coverage bias, i.e. distributional differences between what labeling functions can label and the full data distribution. My guess is that in some cases where the GT (Cov)-GT (Full) gap exists, coverage bias could be closely tied to difficulty, e.g., LFs fail to cover examples close to the ideal decision boundary, leading to a sub-optimal model. There may be other cases though where this link does not hold; perhaps the uncovered examples are not difficult but simply poorly represented by covered ones (i.e., they are “far away” from both the ideal decision boundary and all covered examples). In future work, it would be interesting to investigate/disentangle these scenarios further!
>
> **Questions**
>
> **(1)** Correct, many of the techniques in our design space could be ported over to the setting where there are generic noisy labels. The exception would be anything that involves the label model, such as in the version of self-training where the end model is used as an LF. In the settings typically studied for learning with generic noisy labels, one also isn’t usually given estimated soft-labels to start with, so some other technique would have to be applied to get confidence estimates (i.e., if one wants to use confidence-based thresholding).
>
> **(2)** For Massive18/Banking77, one perspective on why a gap exists is that these are more complex tasks (at least in terms of the label cardinality). The datasets with relatively small label spaces in WRENCH (e.g., Yelp, Imdb, AGNews) are the ones that show smaller gaps between (1) and (2), whereas datasets like Chemprot (|Y|=10) and Semeval (|Y|=9) do show larger gaps. As for the tabular datasets, we suspect that not having a pre-trained model might be an explanation. For text datasets, it may be that pre-trained models/representations can help “smooth” over the gaps in LF-coverage.
>
> **(3)** The main conclusion we wanted to show with this set of plots is that using SSL can add value in different scenarios (e.g., as coverage levels decrease), contrary to what happens on the original WS benchmarks. For this, we saw using simpler Thresh+SSL methods as sufficient though it is certainly possible that Thresh+SSL+Re-label may further improve upon Thresh+SSL. While we don’t expect the high-level conclusions to change, the coverage range in which SSL is helpful could expand as a result. We plan to include such experiments in future revisions.

---

> > ### Comment · Reviewer_vRqg · 2023-08-15
> >
> > Thank you for your response! I think this is great work and I hope it gets accepted.

---

### Official Review · Reviewer_ohhg · 2023-07-10

**Soundness:** 3 good
**Presentation:** 3 good
**Contribution:** 3 good
**Rating:** 6
**Confidence:** 3

**Summary:**

The paper presents a systematic study of how useful SSL is in  weak supervision. Specifically, the authors analyze SSL and weak supervision (WS) along three axes and explore various approaches along each axis. First,  the paper analyzes what to consider as `unlabeled` data for SSL training. The second axis refers to what SSL method to utilize. Finally the paper analyzes various ways of refining the WS weak labels through relabeling. In short, this paper is a large-scale analysis into SSL and WS which tries to understand where the improvements of SSL and WS reported in prior work stem from. All methods are thoroughly tested on 8 classification WS benchmarks.

**Strengths:**

- The presentation of the paper is great. The study in thoroughly detailed and easy to understand.
- The motivation is also excellent. Sheding light on how to achieve an effective WS training setup and analyzing its interaction with SSL is much needed.
- The paper performs the analysis on as many as 8 datasets and performs a thorough ablation study and analyses.
- There are some very interesting findings, such as the fact that SSL does not help too much in various setups. In a way it makes sense that a high coverage for weak labeling means that the unlabeled data may not contain enough information for the model to learn.

**Weaknesses:**

- I believe the search space considered in the paper could be more comprehensive. I appreciate that the authors acknowledge this in the paper (Limitations), however, given that the paper is effectively an in-depth analysis of various ways of combining WS with SSL, I believe that these analyses are needed. Specifically, besides soft label renormalization and contrastive learning, the paper mentions that augmentations are less straightforward for NLP tasks. However, prior work such as UDA [1], MixText [2] or AUM-ST [3] have effectively used augmentations in SSL setups. I am just a bit concerned that the strength of the SSL methods is not adequate. Given the aim of the paper, these methods should have been taken into account.

- Along the same lines, the novelty of the paper seems a bit limited. Once again, these types of in-depth analyses this study tries to address are much needed, but I believe the scale of the study could be a bit larger.

[1] -  Unsupervised Data Augmentation for Consistency Training
[2] - MixText: Linguistically-Informed Interpolation of Hidden Space for Semi-Supervised Text Classification
[3] - Leveraging Training Dynamics and Self-Training for Text Classification

**Questions:**

- Given the LLM craze and their outstanding few-shot capabilities, I would be curious to see how this analysis would look like when having a strong LLM weak labeler. What do you think?

- I think reducing coverage by downsampling is not adequate. The downsampled set will still have the same distribution whereas a real-world limited coverage would indicate different distributions. Can you comment a bit more on how you did the downsampling?

**Limitations:**

The paper discussed the limitations at length. There is no negative societal impact of this work.

---

> ### Author Rebuttal · Authors · 2023-08-10
>
> Thank you for your time and helpful feedback! To respond to your questions and comments:
>
> **Scale of study / Stronger SSL methods.** We agree that it is possible that more sophisticated SSL techniques could lead to a conclusion that SSL is more useful. Based on your feedback, we implemented and ran UDA (using En-De backtranslations) on three of the text classification benchmarks (i.e, Youtube, Trec, AGNews). Overall, the results shown in Table 1 in the pdf (attached to the global response) do not change the conclusions from our paper: using UDA does not improve upon the best methods we previously found, even when combining it with Thresholding + Re-labeling. We’re happy to expand these results to other tasks and will incorporate them into future revisions!
>
> While UDA can outperform other SSL methods on traditional SSL benchmarks, we suspect that our results here are because the SSL settings induced by WS are quite different in that for labeled set L and unlabeled set U: (i) the |L| : |U| ratio is much higher; (ii) L and U are not i.i.d.; (iii) L contains (feature-dependent) noisy labels. Our analysis in Figure 3 (from the submission) perhaps adds another perspective here, showing that the unlabeled data in these tasks may simply not be necessary for learning a strong model: assuming access to ground-truth labels, the final performance changes little whether one trains on the full training set or just the covered examples.
>
> **LLMs in WS.** We note first that there are many settings in which few-shot learning with LLMs may not be as effective. For example, when the task involves private data and/or narrow domain knowledge, LLMs are less likely to have been trained on sufficiently similar data or to be able to capture all the relevant nuances of the problem (e.g. a company's specific policies/ontology for classifying customer intents). Hence, we believe our focus on weak supervision settings where the supervision comes from primarily human-written labeling functions, remains relevant to real world practice.
>
> Nonetheless, we see using LLMs in WS as an interesting setting. One nuance for our work would be that using an LLM directly as a few-shot labeler (by default) does not leave any examples unlabeled. Thus, applying SSL would require some additional thresholding/abstaining mechanism. Meanwhile, LLMs may also allow for several new techniques to perform thresholding (e.g. via prompting an LLM to provide reasoning/confidence along with individual labels) as well as SSL (e.g. generating augmentations for consistency regularization). These would all be interesting to look into in future work!
>
> **Clarification of downsampling.** In our programmatic supervision setups, the distribution is indeed biased according to where the labeling functions label. We totally agree that downsampling a high-coverage WS training set at a *per-data point level* would not adequately capture the biases of actual low-coverage settings. That is why we instead downsample at a *per-LF level*, which reflects the real world process that more bias can be incurred when labeling fewer points with smaller LF sets. We provided more details about the specific subsampling procedures in Appendix D, but are happy to clarify any points in further back-and-forths.

---

> > ### Comment · Reviewer_ohhg · 2023-08-16
> >
> > Thank you for the response, clarifications, and for providing your view on applicability of LLMs in WS! I still believe additional SSL comparisons could further improve the paper. However, given the comparison provided and after reading other reviewers' comments I think this is enough to warrant acceptance.

---

### Author Rebuttal · Authors · 2023-08-10

We thank all of the reviewers for their time and thoughtful feedback! We will respond to each review individually, using this "global response" space to upload our pdf containing the Tables/Figures for new results (which are referenced in our responses).

---

### Decision · Program_Chairs · 2023-09-21

**Decision:**

Accept (poster)

**Comment:**

Reviewers are positive about this paper. The paper is an analysis paper about semi supervised learning and weakly supervision. Reviewers considered the analysis is extensive with good coverage of benchmarks and methods. Reviewers consider that the paper shines a good light on how to use SSL methods for WS and draws some conclusions in the right level of nuance. Moreover, reviewers believe that this paper should be informative to researchers working in both SSL and WS. Hence acceptance is recommended.